# MaGIC: Multi-modality Guided Image Completion

**Hao Wang**[1,2*]    **Yongsheng Yu**[3*]    **Tiejian Luo**[1]    **Heng Fan**[4]    **Libo Zhang**[2†]

[1]School of Computer Science and Technology, University of Chinese Academy of Sciences
[2]Institute of Software, Chinese Academy of Sciences
[3]Department of Computer Science, University of Rochester
[4]Department of Computer Science and Engineering, University of North Texas
wanghao184@mails.ucas.ac.cn; yongsheng.yu@rochester.edu; tjluo@ucas.ac.cn
heng.fan@unt.edu; libo@iscas.ac.cn

## Abstract

Vanilla image completion approaches exhibit sensitivity to large missing regions, attributed to the limited availability of reference information for plausible generation. To mitigate this, existing methods incorporate the extra cue as a guidance for image completion. Despite improvements, these approaches are often restricted to employing a *single modality* (*e.g.*, *segmentation* or *sketch* maps), which lacks scalability in leveraging multi-modality for more plausible completion. In this paper, we propose a novel, simple yet effective method for **M**ulti-mod**a**l **G**uided **I**mage **C**ompletion, dubbed ***MaGIC***, which not only supports a wide range of single modality as the guidance (*e.g.*, *text*, *canny edge*, *sketch*, *segmentation*, *depth*, and *pose*), but also adapts to arbitrarily customized combination of these modalities (*i.e.*, *arbitrary multi-modality*) for image completion. For building MaGIC, we first introduce a modality-specific conditional U-Net (MCU-Net) that injects single-modal signals into a U-Net denoiser for single-modal guided image completion. Then, we devise a consistent modality blending (CMB) method to leverage modality signals encoded in multiple learned MCU-Nets through gradient guidance in latent space. Our CMB is *training-free*, thereby avoiding the cumbersome joint re-training of different modalities, which is the secret of MaGIC to achieve exceptional flexibility in accommodating new modalities for completion. Experiments show the superiority of MaGIC over state-of-the-art methods and its generalization to various completion tasks.

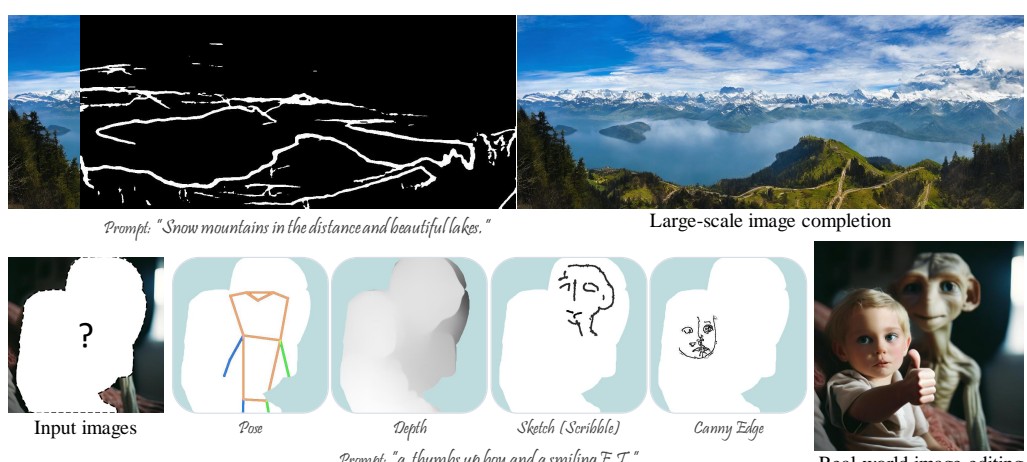

Figure 1: Illustration of our MaGIC for image completion tasks including *outpainting* (first row) and *real user-input editing* (second row) under *multi-modality* guidance.

*Hao Wang and Yongsheng Yu contributed equally to this work.

†Corresponding author: Libo Zhang. This work was supported by Youth Innovation Promotion Association, CAS(2020111).

# 1 INTRODUCTION

Image completion (Criminisi et al., 2003; Li et al., 2022; Lugmayr et al., 2022), involving the concealment of a portion of an image and prompting a model to imaginatively restore it, has long been a subject of extensive research with many applications, such as object removal (Suvorov et al., 2022; Criminisi et al., 2003), image compositing (Levin et al., 2004), photo restoration (Wan et al., 2020), etc. Typical image completion approaches (Li et al., 2022; Suvorov et al., 2022) are prone to struggle with complex or large masking regions due to inadequate reference information. This limitation causes ambiguity to the completion model over restoration or elimination and leads to noticeable artifacts in completed images, degrading the quality.

An intuitive solution to overcome the above limitation is to incorporate user-input (Horita et al., 2022; Yu et al., 2019; Zheng et al., 2022a) or prediction-based (Nazeri et al., 2019; Yu et al., 2022; Guo et al., 2021; Dong et al., 2022) guidance, *e.g.*, text (Avrahami et al., 2023; Xie et al., 2022; Nichol et al., 2022; Wang et al., 2023), edge (Nazeri et al., 2019; Guo et al., 2021; Yu et al., 2022), or segmentation (Yu et al., 2022; Liao et al., 2020; Zheng et al., 2022b), into image completion. However, these approaches are limited to performing image completion under only single-modality guidance, which is *inflexible* in employing the multi-modality, especially more than two modalities simultaneously, for plausible generation and leads to limited application scenarios.

Recently, denoising diffusion probabilistic model (Ho et al., 2020) has been widely employed and demonstrated superior performances in text-to-image synthesis (Rombach et al., 2022; Gu et al., 2022; Ramesh et al., 2022) and text-driven image manipulation fields (Kim et al., 2022; Avrahami et al., 2022; Kawar et al., 2023). In addition to *text*, many approaches (Bansal et al., 2023; Yu et al., 2023; Chen et al., 2023; Avrahami et al., 2022) have explored the integration of extra guidance modality, such as *segmentation*, *sketch*, *pose*, and even *position* of generated object, into diffusion models in a *training-free* way. These methods involve designing energy loss associated with the input guidance and guiding its gradient on the latent codes during inference, yet they tend to fail to maintain fine-grained structural information, resulting in insufficient control over the generated results. Meanwhile, several *training-required* approaches (Mou et al., 2023; Zhang & Agrawala, 2023) have further enhanced the control of input modality over diffusion models by introducing an auxiliary conditional network to encode modality and directly add the encoded features to the intermediate features of frozen diffusion models. These methods bring in fresh insights and pave the way for incorporating guidance signals into image completion. Nevertheless, simply transferring these ideas to multi-modality image completion is *not trivial*, as the introduction of each new modality necessitates the joint training of all auxiliary conditional networks. How to effectively integrate multi-modality guidance for image completion in a *scalable* and *flexible* manner remains an open problem.

In this paper, we propose *MaGIC*, a *novel*, *simple* yet *effective* framework for **M**ulti-mod**a**lity **G**uided **I**mage **C**ompletion, especially when there are more than two modalities at the same time. MaGIC is designed to be *scalable* and *flexible*, allowing it to merge various modalities, including but not limited to *text*, *canny edge*, *sketch*, *segmentation*, *depth*, and *pose*, in an arbitrary combination as guidance for image completion (see Fig. 1 and Fig. 2). To build MaGIC, there are two core ingredients, including a *modality-specific conditional U-Net* (MCU-Net) and a *consistent modality blending* (CMB) method, performed in two stages.

Specifically, the proposed MCU-Net, composed of a standard U-Net denoiser from the pre-trained stable diffusion (Rombach et al., 2022) and a simple encoding network, which injects a single modality guidance signal into the U-Net denoiser to attain single-modal guided completion. The MCU-Net will be individually finetuned under each single modality, in the first stage. Then, to achieve multi-modality guidance, the CMB algorithm is proposed in the second stage to flexibly aggregate guidance signals from any combination of previously learned MCU-Nets. The CMB leverages guidance loss to gradually narrow the distances between the intermediate features from the original pretrained U-Net denoiser and multiple MCU-Nets during the denoising sample stage, which ensures that the former features do not deviate too much from the original feature distribution during multi-modality guidance. Compared with the naive approach of achieving multi-modality guided completion by jointly re-training a unified model, our CMB is *training-free* and allows for the flexible addition or removal of guidance modalities, avoiding cumbersome re-training and preserving the feature distribution of the original U-Net denoiser.

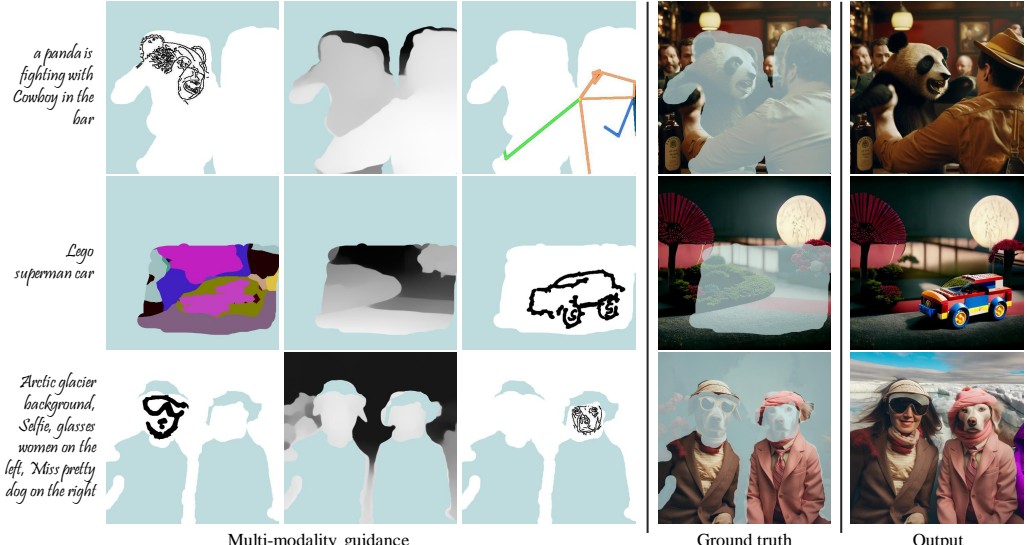

Figure 2: Illustration of our MaGIC for real user-input editing task using various combination of *multi-modality* as guidance.

To verify the proposed MaGIC, we conduct extensive experiments on various tasks including image inpainting, outpainting, and real user-input editing, using the COCO (Lin et al., 2014), Places2 (Zhou et al., 2018), and in-the-wild data. Our results demonstrate the superiority of MaGIC over image completion and controllable generation baselines in terms of image quality. In addition, we find that, surprisingly, the CMB of our MaGIC is also well applicable for multi-modality guided image generation, showing its generality and potential for generative tasks. Fig. 3 illustrates the architecture of our approach.

In summary, our **contributions** are four-fold: **(i)** we propose a novel approach of MaGIC for flexible and scalable multi-modality guided image completion. To the best of our knowledge, MaGIC is the first to widely support arbitrary multi-modality guided image completion; **(ii)** we present a simple yet effective MCU-Net to effectively and adaptively inject a modality as guidance for image completion; **(iii)** we introduce a novel CMB algorithm that combines arbitrary multiple modalities for image completion without the need for additional training and **(iv)** using MaGIC, we achieve performance superior to that of other state-of-the-art approaches.

## 2 RELATED WORK

**Auxiliary-based image completion.** The auxiliary-based image completion methods aim to enhance the structure and texture of completed images by incorporating predicted or human-provided prior information. Early approaches primarily focus on using a single modality (*e.g.*, edge (Nazeri et al., 2019; Guo et al., 2021; Dong et al., 2022; Zheng et al., 2022a) or segmentation (Zheng et al., 2022b; Liao et al., 2020)) as the auxiliary guidance for image completion. Recently, inspired by the superior-performing diffusion models (Ho et al., 2020; Dhariwal & Nichol, 2021; Rombach et al., 2022), text-based auxiliary solutions have been proposed for image completion (Wang et al., 2023; Avrahami et al., 2023; Nichol et al., 2022; Avrahami et al., 2022), providing more user-friendly image editing applications.

However, prompt text alone is not sufficient. Because the above methods are constrained by the training requirements of auxiliary guidance, it is difficult to flexibly add more types of modalities as guidance for completion. Our MaGIC can incorporate arbitrary combination of multiple modalities for more plausible completion results (see Fig. 1 again). It is versatile, requiring only the optimization of single-modality conditional networks, and allows for plug-and-play integration into the conditional image completion process without the need for additional cumbersome joint re-training.

**Controllable image generation with diffusion models.** Diffusion models (Ho et al., 2020; Dhariwal & Nichol, 2021; Rombach et al., 2022; Song et al., 2023) have drawn extensive attention in image generation owing to their remarkable results and stable training. These methods can be broadly categorized into *train-required* and *train-free* approaches. The former achieves powerful generation control by training on large-scale data or fine-tuning a conditional control sub-network on pre-trained diffusion models (*e.g.*, (Rombach et al., 2022)). Recent research (Zhang & Agrawala, 2023; Mou et al., 2023) has introduced various modalities (*e.g.*, keypose point maps, sketch maps, etc) for generation. However, it *fails* to simultaneously use multi-modality as guidance. Differently, train-free solutions (Yu et al., 2023; Chen et al., 2023; Bansal et al., 2023; Jeong et al., 2023) leverage the multi-step nature of diffusion models, explicitly introducing guidance signals during the iterative denoising process and achieving style (Jeong et al., 2023), layout (Chen et al., 2023; Bansal et al., 2023), face identity (Bansal et al., 2023; Yu et al., 2023), segmentation map (Bansal et al., 2023; Yu et al., 2023) guidance without fine-tuning. Yet, they struggle to leverage fine-grained structural guidance (*e.g.*, canny edge) as conditions, potentially resulting in degraded guidance (Yu et al., 2023).

Our MaGIC is inspired by the above image generation approaches, but different in two aspects. First, MaGIC achieves multi-modality guidance without joint re-training while improving the effectiveness of fine-grained structure guidance. In addition, MaGIC goes beyond controllable generation and can be applied to guided completion and real-world editing tasks.

## 3 MaGIC: Multi-modality Guided Image Completion

Masked images $x_m = x \odot m$ are obtained by corrupting images $x$ with binary masks $m \in \{0, 1\}^{H \times W \times 1}$, where $x \in \mathbb{R}^{H \times W \times 3}$ are original RGB images with width $W$ and height $H$. Given a known region $x_{\overline{m}} = x \odot (1 - m)$, the goal of image completion is to learn a function $p(x_m | x_{\overline{m}})$ that completes the missing mask area with visually realistic and structurally coherent content. To mitigate the inherent ambiguity of completion model, the direction of restoration or elimination is controlled through the auxiliary guidance $C$. In the following sections, we start by outlining necessary diffusion steps in 3.1) for formulating our method, then elaborate on MaGIC, addressing auxiliary guidance via our proposed MCU-Net in 3.2 and multi-modality integration by our CMB algorithm in 3.3.

### 3.1 Preliminaries

**Diffusion models.** Denoising diffusion probabilistic models (DDPMs) (Ho et al., 2020) are generative models that learn the true distribution $p(x_T)$ by iteratively denoising a randomly sampled noise image $x_T$. In each denoising step, a U-Net model is trained to predict the noise $\epsilon$ based on the objective function,

$$\Phi(x_t, t, \theta) = \min(\mathbb{E}_{x_0, t, \epsilon \sim \mathcal{N}(0,I)} \| \epsilon - \epsilon_\theta^t(x_t) \|_2^2), \tag{1}$$

where $x_t = \sqrt{\alpha_t} x_0 + \sqrt{1 - \alpha_t} \epsilon$ represents the intermediate noised image obtained after applying noise $t$ times to the clean image $x_0$, and $\alpha_t = \prod_{s=1}^{t}(1 - \beta_s)$ is a series of fixed hyperparameters based on the variance schedule $\beta_s, s \in [1, T]$. The model can be further generalized to conditional generation (Dhariwal & Nichol, 2021; Ho & Salimans, 2021), with predicted noise becoming $\epsilon_\theta(x_t, t, C)$.

**Stable diffusion.** We consider stable diffusion (SD) inpainting model (Rombach et al., 2022) as the main backbone in the subsequent method sections. Instead of beginning with isotropic Gaussian noise samples in pixel space, the SD model first maps clean images to their corresponding latent space $\mathcal{Z}$ through $E(\cdot)$. Here, $E(\cdot)$ is an autoencoder with a left inverse $D$, ensuring $x = D \circ E(x)$. Owing to the lower inference overhead of U-Net in the latent space, SD has emerged as an important class of recent image generators based on diffusion (Rombach et al., 2022; Saharia et al., 2022; Zhang & Agrawala, 2023; Avrahami et al., 2023). Specifically, the initial latent codes of iterative denoising process employ random $z_T \sim \mathcal{Z} \in \mathbb{R}^{\frac{H}{s} \times \frac{W}{s} \times 3}$, where $s$ signifies $s$-fold reduction in spatial dimensions. The mask and encoding masked image serve as conditions for the U-Net, modifying the objective function in Eq. 1 to

$$\Phi(z_t, t, m_\downarrow, x_{m\downarrow}, \theta) = \min(\mathbb{E}_{z_0, t, \epsilon \sim \mathcal{N}(0,I)} \| \epsilon - \epsilon_\theta^t(z_t, m_\downarrow, x_{m\downarrow}) \|_2^2), \tag{2}$$

where $m_\downarrow \in \mathbb{R}^{\frac{H}{s} \times \frac{W}{s} \times 1}$ denotes the $s$-fold nearest-neighbor downsampling of the input mask $m$, and $x_{m\downarrow} = E(x_m)$ indicates embedded masked image in latent space. Denoising diffusion implicit model

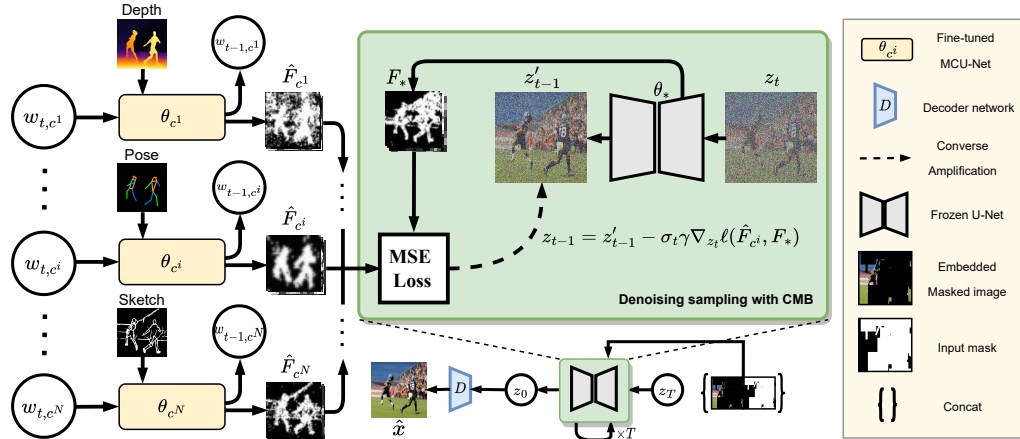

Figure 3: Illustration of our method. We initiate the inference process with a randomly initialized latent $z_T$. This latent is denoised $T$ times, with the concatenation of the masked image and mask acting as conditioning for both *MCU-Net* and frozen U-Net denoiser. Through *CMB*, we fuse diverse modality guidance signals, aiding the frozen original U-Net $\theta_*$ to iteratively produce the desired content. The content is finally transformed into pixel space via a decoder network, resulting in the completed RGB output.

(DDIM) (Song et al., 2021) defines each step of denoising as a non-Markovian process while retaining the same training objective as DDPM. Accordingly, the sampling process is formulated as,

$$z_{t-1} = \sqrt{\alpha_{t-1}}\left(\frac{z_t - \sqrt{1-\alpha_t}\epsilon_\theta^t(z_t, m_\downarrow, x_{m\downarrow})}{\sqrt{\alpha_t}}\right) + \sqrt{1-\alpha_{t-1}-\sigma_t^2} \cdot \epsilon_\theta^t(z_t, m_\downarrow, x_{m\downarrow}) + \sigma_t\epsilon_t, \quad (3)$$

where the noise $\epsilon_t$ follows the standard normal distribution $\mathcal{N}(0, \mathbf{I})$ and is independent of $x_t$, and $\sigma_t = \eta\sqrt{(1-\alpha_{t-1})/(1-\alpha_t)}\sqrt{1-\alpha_t/\alpha_{t-1}}$. By gradually denoising over $T$ timesteps, the content of the missing region is hallucinated in the latent space, producing a conditional sample $z_0 \sim p(z_T|m_\downarrow, x_{m\downarrow})$. $z_0$ is then transformed into the pixel space as $\hat{x} = D(z_0)$ via the left-inverse decoder network $D$ corresponding to the autoencoder $E(\cdot)$, finally resulting in the completion outcome $\hat{x} \in \mathbb{R}^{H \times W \times 3}$.

## 3.2 MCU-Net: Modality-specific Conditional U-Net

The first stage in MaGIC is to learn image completion under single-modality guidance. For this purpose, we propose a simple yet effective modality-specific conditional U-Net (MCU-Net). Particularly, for the auxiliary guidance $c^i \in C$ ($C = \{c^i\}_{i=1}^N$ denotes the set of $N$ auxiliary guidance), MCU-Net consists of a standard U-Net denoiser $\theta_{c^i}$ (Rombach et al., 2022) and an encoding network $\tau_{c^i}$. For simplicity, we will omit $i$ in the following sections.

The encoding network $\tau_c$ is employed to extract multi-scale guidance signals, represented as $F_c^l$, where $l \in \{0, \cdots, L\}$ and $L$ denotes the number of times the feature map scale is reduced within the U-Net denoiser. Afterwards, $F_c^l$ is injected to the latent in MCU-Net to obtain modality-guided feature. In specific, we denote the latent in MCU-Net as $w_{t,c}$ ($c \in C$) to distinguish it from the original diffusion model's $z_t$. As illustrated in Fig. 4, to inject guidance signals

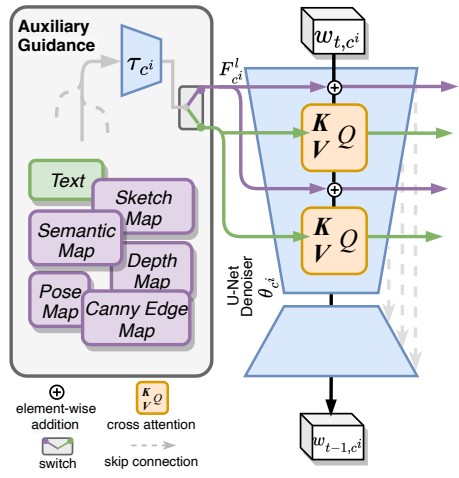

Figure 4: Illustration of MCU-Net.

into the latent $w_{t,c}$, we add $F_c^l$ to intermediate feature maps $F_{enc}^l$ of the encoder of MCU-Net, resulting in guided feature map $\hat{F}_c^l = F_{enc}^l + F_c^l, l \in [0, L]$. And we incorporate the text modality in a manner

consistent with SD, which integrates its information into intermediate features via a cross-attention mechanism.

To utilize the generative capability of pre-trained SD, we freeze the original U-Net denoiser when training MCU-Net, allowing the unlocked encoding network $\tau_c$ to learn guidance signal extraction and fit the pre-trained denoiser.

### 3.3 CMB: CONSISTENT MODALITY BLENDING

Despite achieving image completion under single-modality with MCU-Net, it is *not trivial* to integrate multiple MCU-Nets for multi-modality image completion. A naive way is to jointly re-train these learned MCU-Nets, which is cumbersome and inflexible for multi-modality image completion. To deal with this, we propose the novel consistent modality blending (CMB), a *training-free* algorithm to integrate guidance signals from different auxiliary modalities *without* requiring additional joint re-training. A great benefit of CMB is that, the multi-modality guidance latent code in MCU-Net remains aligned with the internal knowledge of SD model, without affecting its original ability. As shown in Fig. 3, the guidance signals from arbitrary combination of independent single-modality models (*i.e.*, MCU-Nets) in gradient aspect gradually control the image completion process with input modalities.

Specifically, given a series of MCU-Nets trained independently on multiple modalities $C$, we can extract the guidance signals $F_c$. A simple way for integrating different modalities is to directly update intermediate feature maps $F_{enc}$ by adding accumulated guidance signals as $\hat{F}_C \leftarrow F_{enc} + \sum_{c \in C} F_c$. However, we argue that this simple manner we called *feature-level addition* is impractical, as the denoiser is trained solely on the distribution of $\hat{F}_c = F_{enc} + F_c$. Drawing inspiration from recent advancements in classifier-guidance diffusion (Dhariwal & Nichol, 2021), we introduce a converse amplification strategy. This technique enables the intermediate feature maps $F_*$ of an original U-Net to more closely approximate each guided feature map $\hat{F}_c$.

**Converse Amplification**. We use $F_*$ to denote the intermediate features from the original U-Net $\theta_*$ which is not equipped with a guidance encoding network, while $\hat{F}_c$ to denote guided features from MCU-Net $\theta_c$ of modality $c$. Notably, U-Net $\theta_*$ and MCU-Net $\theta_c$ undergo a parallel denoising process. At each step $t$, every latent is denoised using the DDIM sampler (Song et al., 2021). In the original U-Net $\theta_*$, we denote the denoised latent as the intermediate latent $z'_{t-1}$.

We bias $F_*$ towards $\hat{F}_c$ by calculating their Euclidean distance in each scale $l$:

$$\ell(\hat{F}_C, F_*) = \frac{1}{L} \sum_{l=0}^{L} \sum_{c \in C} \delta_c \|\hat{F}_c^l - F_*^l\|_2^2, \quad (4)$$

where $\delta_c$ are scale factors to weight the strength leads to either improved alignment to guidance modality $c$ or greater diversity in the outputs. $N = |C|$ indicates the modality number of auxiliary guidance set. We then apply the distance as an energy loss, similar to the *classifier guidance* methods in Chen et al. (2023); Dhariwal &

---

**Algorithm 1** Usage of CMB in MaGIC

**Require:** Given the input masked image $x_m$, mask $m$, a series of MCU-Net parameters $\theta_c$, the number of times of converse amplification $P$, and the number of iteration steps of back-propagation $Q$.

1: $m_\downarrow = \mathrm{downsample}(m)$
2: $x_{m\downarrow} = E(x_m)$
3: $z_T \sim \mathcal{N}(0, \mathbf{I})$
4: $w_{T,c} \sim \mathcal{N}(0, \mathbf{I}), \forall c \in C$
5: **for** $t = T, \cdots, 1$ **do**
6:     **if** $t \leq T - P$ **then**
7:         $\epsilon_{\theta^*}, F_* \leftarrow \theta_*(z_t, t, m_\downarrow, x_{m\downarrow})$
8:         $z_{t-1} = \mathrm{sampler}(z_t, \epsilon_{\theta^*})$    (Eq. 3)
9:         continue
10:     **end if**
11:     **for** $1, \cdots, Q$ **do**
12:         $\epsilon_\theta, \hat{F}_C \leftarrow \theta_C(w_{t,C}, t, m_\downarrow, x_{m\downarrow})$
13:         $w_{t-1,C} = \mathrm{sampler}(w_{t,C}, \epsilon_\theta)$    (Eq. 3)
14:         $\epsilon_{\theta^*}, F_* \leftarrow \theta_*(z_t, t, m_\downarrow, x_{m\downarrow})$
15:         $z'_{t-1} = \mathrm{sampler}(z_t, \epsilon_{\theta^*})$    (Eq. 3)
16:         $z_{t-1} = z'_{t-1} - \sigma_t \gamma \nabla_{z_t} \ell(\hat{F}_C, F_*)$    (Eq. 5)
17:     **end for**
18: **end for**
19: **return** $D(z_0)$

---

Nichol (2021), to adjust latent code of the original SD model. Specifically, at each denoising step, we obtain $\hat{F}_C$ and $F_*$ firstly, then the gradient of their distance is calculated through back-propagation to update the denoised latent $z'_{t-1}$:

$$z_{t-1} = z'_{t-1} - \sigma_t \gamma \nabla_{z_t} \ell(\hat{F}_C, F_*) \tag{5}$$

Owing to CMB, it is *not necessary* to jointly re-train the learned MCU-Nets, making MaGIC flexible in merging arbitrary multi-modality for completion. Alg. 1 shows the procedure of CMB for MaGIC.

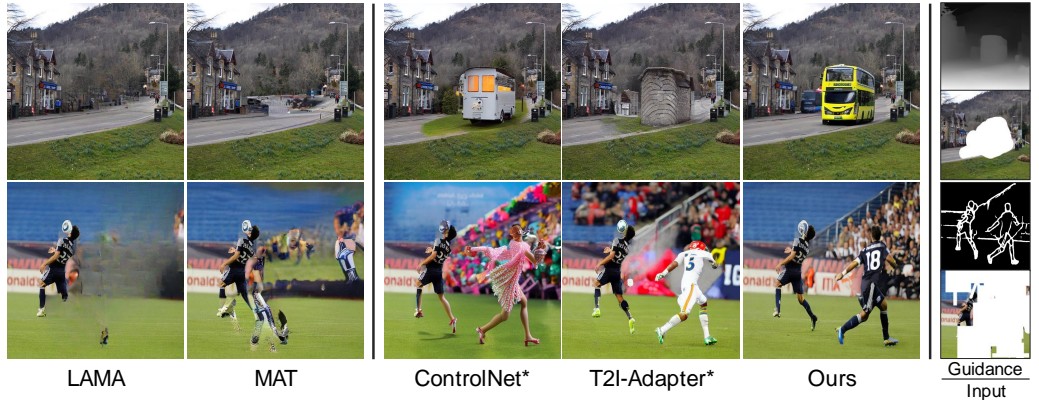

LAMA     MAT     ControlNet*     T2I-Adapter*     Ours     Guidance / Input

Figure 5: Qualitative comparison for image completion using *single* modality as guidance. * indicates the use of latent-level blending (Avrahami et al., 2023) to preserve pixels in unmasked regions.

## 4 EXPERIMENTS

In this work, we study three research questions, **RQ1**, **RQ2** and **RQ3**:

**RQ1:** *Can our MCU-Net effectively perform image completion guided by various modalities?*
**RQ2:** *Can our MaGIC with CMB seamlessly integrate guidance from multiple modalities to produce credible completion results?*
**RQ3:** *How do different module designs (e.g., adjustments in hyperparameters and inference processes) impact the overall effectiveness?*

### 4.1 EXPERIMENTAL SETTINGS

In our experiments, we select several edge-based image completion methods, including EC (Nazeri et al., 2019), CTSDG (Guo et al., 2021), ZITS (Dong et al., 2022), and state-of-the-art (SOTA) techniques such as LAMA (Suvorov et al., 2022), LDM (Rombach et al., 2022), and MAT (Li et al., 2022). We also include controllable image generation baselines such as ControlNet (Zhang & Agrawala, 2023) and T2I-Adapter (Mou et al., 2023) in our qualitative comparison, as they can be easily adapted to the image completion task with the concept of Blended Diffusion (Avrahami et al., 2022; 2023). For fair comparison, we apply the same set of image mask pairs across all tests, and, for comparisons involving auxiliary guidance, we ensure that each method receives identical guidance map instructions. The masks used in testing are designed to uniformly span a masking ratio range from 0 to 100%. The evaluation adopts both image metrics (*i.e.*, FID and P/U-IDS (Zhao et al., 2021)) and text-to-image metric (*i.e.*, PickScore (Kirstain et al., 2023)) which gauges the fidelity of generated content based on learned human preferences. Acknowledging the pluralistic outcomes of our method, we conduct tests on a total of five images to determine mean scores and standard deviations. For all diffusion-based methods, the denoising step $T$ is set to 50. For further details on the experimental configuration, please see the supplementary material.

### 4.2 IMAGE COMPLETION WITH SINGLE-MODALITY GUIDANCE USING MCU-NET

To answer **RQ1**, we compare our approach with state-of-the-art (SOTA) inpainting methods (Suvorov et al., 2022; Li et al., 2022) and SOTA single modality guidance image generation methods. We employ latent-level blending (Avrahami et al., 2023) to preserve pixels in unmasked regions for image generation methods such as ControlNet (Zhang & Agrawala, 2023) and T2I-Adapter (Mou et al., 2023). As depicted in Fig. 5, our method generates content without noticeable artifacts, maintaining stronger spatial context consistency. Conversely, T2I-Adapter generates a stone house on the road (1st row in Fig. 5) and ControlNet puts a dancer on the soccer field (2nd row in Fig. 5).

Quantitatively, the scores of edge-based methods on COCO and Places2 are displayed in Tab. 1. Across all metrics, our method demonstrates significant improvements, indicating that our MCU-Net can effectively generate content under the guidance of various single modalities.

| Method | COCO | | Places2 | | |
| --- | --- | --- | --- | --- | --- |
| | FID↓ | PickScore↑ / % | FID↓ | U-IDS↑ / % | P-IDS↑ / % |
| EC (Nazeri et al., 2019) ♠ | 76.64 | 23.14 | 25.08 | 12.89 | 2.86 |
| CTSDG (Guo et al., 2021) ♠ | 97.05 | 24.03 | 42.81 | 0 | 0 |
| ZITS (Dong et al., 2022) ♠ | 61.27 | 28.09 | 18.96 | 18.75 | 7.20 |
| Our MCU-Net† | 47.70±0.29 | 30.79±0.10 | 10.74±0.07 | 23.83±0.30 | 10.18±0.48 |
| Our MCU-Net ♥ | **39.43±0.26** | **37.12±0.11** | 9.09±0.04 | 25.34±0.29 | 10.64±0.46 |
| Our MCU-Net ♣ | 41.91±0.20 | 34.96±0.17 | 10.27±0.06 | 24.21±0.24 | 9.93±0.38 |
| Our MCU-Net ♠ | 41.15±0.27 | 34.94±0.06 | **8.32±0.02** | **26.23±0.07** | **10.96±0.33** |

Table 1: Comparison of using single auxiliary modality as guidance for image completion. ♠: ground truth edge map as guidance, ♥: estimated depth map as guidance, ♣: segmentation map as guidance, ↑: the higher the better, ↓: the lower the better, †: completion without any guidance.

| Method | MMG | COCO | |
| --- | --- | --- | --- |
| | | FID ↓ | PickScore ↑ / % |
| MaGIC w/ FLA (35 steps) | ✓ | 37.78±0.32 | 44.19±0.23 |
| MaGIC w/ FLA (50 steps) | ✓ | 41.53±0.19 | 35.85±0.08 |
| MaGIC† | ✗ | 47.70±0.29 | 30.79±0.10 |
| MaGIC w/ CMB | ✓ | **37.65±0.22** | **49.57±0.17** |

(a) Comparison of CMB with simple FLA.

| Method | MMG | COCO | |
| --- | --- | --- | --- |
| | | FID ↓ | PickScore ↑ / % |
| CoMod | ✗ | 68.01 | 25.12 |
| TFill | ✗ | 58.55 | 24.63 |
| FcF | ✗ | 48.92 | 26.43 |
| LAMA | ✗ | 48.63 | 29.06 |
| MAT | ✗ | 45.51 | 27.10 |
| MaGIC† | ✗ | 47.70±0.29 | 30.79±0.10 |
| MaGIC | ✓ | **37.65±0.22** | **49.57±0.17** |

(b) Comparison of MaGIC with SOTA methods.

Table 2: Comparisons of CMB and FLA and MaGIC with others. MMG: multi-modality guidance.

### 4.3 IMAGE COMPLETION WITH MULTI-MODALITY GUIDANCE USING MaGIC

We further decompose **RQ2** into two smaller questions, **RQ2.1** (*Is CMB effective?*) and **RQ2.2** (*How does MaGIC perform?*)

**Answering RQ2.1.** CMB aims to integrate different modalities as guidance for image completion in a *training-free* fashion. Compared with CMB, a simple way is to aggregate feature maps $F_c$ ($c \in C$) by addition (*i.e.*, *feature-level addition* or *FLA* for short) to produce $\hat{F}_C$ as $\hat{F}_C \leftarrow F_{enc} + \sum_{c \in C} F_c$. To show the effectiveness of CMB, we compare it with FLA on COCO as in Tab. 2a. Note that, we test FLA with 30 and 50 steps, respectively. To guarantee an equitable assessment across all auxiliary modalities, we opt for a wide-ranging set of modalities. Given that specific modality (*e.g.*, pose) may not be applicable to all test images (*e.g.*, certain landscape images), we ensure that our test suite incorporates a diverse range of modalities. This includes segmentation map, depth map, Canny edge map, sketch map, and a prompt text. As displayed in Tab. 2a, the proposed CMB significantly surpasses FLA with naive addition, evidencing the effectiveness of CMB in merging multi-modality for completion. Interestingly, the performance of FLA with 50 steps is counter-intuitively lower than that with 35 steps, suggesting that this simple method may overly manipulate the latent code. This indicates that the direct addition of different MCU-Net feature maps for multi-modality guidance is *impractical*. By contrast, our CMB efficaciously integrates the signals from multi-modal guidance.

**Answering RQ2.2.** To validate the effectiveness of our MaGIC, we compare it with state-of-the-art image completion methods, including LAMA (Suvorov et al., 2022) and MAT (Li et al., 2022), on COCO. As in Tab. 2b, our guidance-free inpainting model denoted as MaGIC† is comparable to SOTA inpainting baselines MAT and LAMA. When employing multi-modality (segmentation, canny edge, sketch, depth, and text) as guidance, our MaGIC gains significant improvements. In comparison to MaGIC†, we obtain gains of 21% in FID and 61% in PickScore. Notably, the PickScore implies that, from the perspective of learned human preference, our completed images have a 49.57% chance of being more faithful to the ground truth image caption than the original images.

### 4.4 ABLATION STUDY

To answer **RQ3**, we conduct rich ablations on COCO as follows.

| text | seg | depth | canny | sketch | FID ↓ | PickScore ↑ / % |
|---|---|---|---|---|---|---|
| ✓ | ✓ | ✓ | ✓ | ✓ | 35.42±0.10 | 47.81±0.34 |
|  | ✓ |  |  |  | 41.91±0.20 | 34.96±0.17 |
|  |  | ✓ |  |  | 39.43±0.26 | 37.12±0.11 |
|  |  |  | ✓ |  | 41.15±0.27 | 34.94±0.06 |
|  |  | ✓ | ✓ |  | 41.38±0.22 | 35.99±0.13 |
|  |  |  | ✓ | ✓ | 42.59±0.19 | 34.98±0.16 |
| ✓ |  |  |  |  | 39.96±0.12 | 48.58±0.25 |
| ✓ | ✓ | ✓ | ✓ | ✓ | 37.65±0.22 | 49.57±0.17 |

| $P$ | $Q$ | FID↓ | PickScore↑ / % |
|---|---|---|---|
| 30 | 1 | 38.50±0.23 | 48.11±0.21 |
| 30 | 10 | 38.20±0.28 | 48.60±0.12 |
| 10 | 5 | 40.40±0.25 | 45.98±0.12 |
| 20 | 5 | 38.37±0.34 | 48.72±0.15 |
| 30 | 5 | 37.65±0.22 | 49.57±0.17 |
| 40 | 5 | 36.78±0.22 | 50.64±0.20 |
| 50 | 5 | 36.60±0.12 | 50.52±0.07 |

(a) Ablation on different modalities.
(b) Hyperparameter analysis of CMB.

Table 3: Ablation studies on the multi-modality complementary and the hyper-parameters of CMB.

**Impact of modalities.** To delve into auxiliary modalities, we investigate their individual contributions. We distinguish among five modalities used in our experiments: edge and sketch for fine-grained structural control, segmentation and depth for coarse-grained spatial-semantic control, and text for content-specific cue. As in Tab. 3a, the guidance from text significantly enhances image quality (FID) and generated content (PickScore). Interestingly, excluding text, the performance of combined modalities appears balanced, suggesting optimal generation quality when modalities provide complementary information. When using all modalities, the performance is the best.

**Joint multi-modality re-training.** Our method allows multi-modality guidance without the need for additional joint training. However, exploring the joint re-training of all modality-specific conditional U-Nets with classifier-free guidance style can help identify the upper bound performance.

Building such a model necessitates a fuser mechanism to blend diverse input modalities. To ensure effectiveness, we integrated CoAdapterFuser (Mou et al., 2023), aligning with our design goals. Addressing the lack of a paired dataset with extensive labels across various modalities was also essential. We extracted 650,000 images from the Laion dataset and generated four modalities (canny edge map, depth map, sketch map, and semantic map) using open-source tools. During joint re-training, we randomly dropped out each modality at a 0.5 probability. This training process is memory-intensive, necessitating a reduction in batch size to a quarter of single-modality training. The model underwent 180,000 iterations. As evidenced by its lower FID, shown in the first row of Tab. 3a, the unified model achieves higher fidelity than our training-free method. However, it encounters issues such as the need for paired training data, difficulty in adding new modalities, and substantial computational requirements for joint training.

**Guidance in iteration**. The proposed CMB algorithm involves two important hyperparameters, *i.e.*, the number $P$ of denoising steps incorporating CMB and the iteration times $Q$ of gradient descent performed in each CMB operation. We study the impact of different $P$ and $Q$ on the multimodal conditioning completion task as in Tab. 3b. From Tab. 3b, we can observe that with $Q$ fixed, the performance is almost consistently improved by increasing the number $P$ of denoising steps (from 10 to 50) equipped for CMB. Interestingly, given the fact in Tab. 2a that incorporating guidance through simple FLA could impair the performance of completion, the results further demonstrate the effectiveness of CMB.

## 5 Conclusion and Limitation

In this paper, we propose a novel, simple yet effective method, named MaGIC, for multi-modality image completion. Specifically, we first introduce the MCU-Net that is used to achieve single-modality image completion by injecting the modality signal. Then, we devise a novel CMB algorithm that integrates multi-modality for more plausible image completion. On extensive experiments, we show that MaGIC shows superior performance. Moreover, it is generally applicable to various image completion tasks such as in/out-painting and local editing, and even the image generation task.

MaGIC is proposed to facilitate image completion with multi-modality. Yet, there exist two limitations. First, the ability to generate high-frequency details is tied to the backbone completion model, which means even with ample detailed guidance, achieving the desired fidelity may not be guaranteed. This can be improved by adopting more powerful backbones if necessary. In addition, our MaGIC is less efficient than current single-step completion models, with inference time increasing in line with guidance modalities. This is a common issue for diffusion models, and we leave it for future research.

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

# MaGIC: Multi-modality Guided Image Completion

# ————Appendix————

CONTENTS

# A    IMPLEMENTATION DETAILS

## A.1    DIFFERENT IMAGE-BASED CONDITIONS AND HYPER-PARAMETERS

Our experiments include 6 types of image-based conditions:

- *Canny edge & Sketch.* We utilize the training set of COCO (Lin et al., 2014), which contains $123K$ images, as the training data to train MCU-Net separately under canny and sketch guidance. The corresponding canny edge and sketch are generated by Canny algorithm (Canny, 1986) with default thresholds, and PiDiNet (Su et al., 2021) with a threshold of 0.5, respectively.

- *Segmentation.* We utilize training set of COCO-Stuff (Caesar et al., 2018) as training data, which includes $123K$ images and corresponding semantic segmentation annotations. It covers 80 thing classes, 91 stuff classes and 1 "unlabeled" class, providing a comprehensive range of semantic information for MCU-Net training.

- *Depth.* In order to obtain sufficient volume of data to train MCU-Net under this conditions with abstract representation, we select $650K$ images from LAION-AESTHETICS dataset (Schuhmann et al., 2022). And we adopt MiDaS (Ranftl et al., 2022) on them to generate depth maps.

- *Pose.* We also pick images from LAION-AESTHETICS (Schuhmann et al., 2022) to construct training data for MCU-Net under pose guidance. The key distinction from building training dataset for depth guidance is that the selected images must contain at least one person for pose generation. To achieve this, we employ MM-Pose (Contributors, 2020), an open-source toolbox for pose estimation, to filter out images that do not meet the requirement, and generate pose for the retained images. In the end, we gather a total of $600k$ image-pose pairs to train MCU-Net under this condition.

- *Text.* Within our default backbone, the *SD-2.1 Inpainting*, the prompt text is conditioned as the key and value of the cross-attention mechanism in the U-Net denoiser. It's noteworthy that this backbone is pretrained with the prompt text in the classifier-free way (Ho & Salimans, 2021). Consequently, in this work, we opt to use the backbone directly, thus bypassing the necessity to fine-tune an MCU-Net for text guidance.

All our experiments are conducted using 8 NVIDIA A100-40G GPUs. We set the batch size to 64 and employed the Adam optimizer (Kingma & Ba, 2015) with the learning rate of 1e-5 for training 10 epochs. These settings remain consistent across all conditions.

## A.2    ACQUISITION OF CONDITIONS

To facilitate a reliable and convenient comparison of model performance, we employed the conditions provided by the dataset directly or leveraged existing tools (Yang et al., 2022; Contributors, 2020; Ranftl et al., 2022) to estimate them. We then evaluated the model performances using quantitative metrics on completing the corresponding masked RGB images. It is important to note that our method also supports the input of manually designed guidance conditions (as shown in Fig. 1, Fig. 15, Fig. 16 and Fig. 17). However, when manually design dense guidance conditions like segmentation and depth maps, it's crucial to ensure their consistency with the information retained in the unmasked regions, particularly in the case of depth maps where values represent the distance between pixels and the camera. Fortunately, sparse conditions like sketch or pose maps can offer sufficient guidance information. We intend to release our code for condition generation, enabling users to obtain modalities including sketch, pose and segmentation maps effortlessly for image editing purposes.

## A.3    ARCHITECTURE OF ENCODING NETWORK $\tau_{c_i}$

The condition encoding network is designed to be simple and lightweight, and serves the purpose of extracting the multi-scale guidance signals from the input condition image. These guidance signals are aligned in size with the intermediate feature maps of the MCU-Net's encoder. As this is not the main focus of our work, we have referred to the design of T2I-Adapter (Mou et al., 2023). Specifically,

it consists of four feature extraction blocks with a downsample module placed between each pair of adjacent blocks, and each feature extraction block is composed of one convolution layer and two residual blocks. Notably, we cannot simply use the denoiser of T2I-Adapter as our backbone for single modality guided image completion. Our work involves gathering multi-modality datasets and training MCU-Net with masked images via Classifier-free guidance loss as follows,

$$\arg \min_{\theta_c} \mathbb{E}_{z_0,t,\epsilon \sim \mathcal{N}(0,I)} \| \epsilon - \epsilon_\theta^t(z_t, m_\downarrow, x_{m\downarrow}, c) \|_2^2. \tag{6}$$

This training enables our model to adeptly fill masked regions while ensuring spatial consistency with the unmasked areas, as shown in Figure 10 and 11. This finding aligns with the observations in Figure 5 of Section 4.2.

## A.4  Experimental Setting

In order to evaluate all baselines and our proposed method in a fair manner, the same image-mask pairs are used in quantitative experiments. Additionally, testing mask samples are obtained based on a uniform distribution ranging from 0% to 100% to encompass the majority of mask ratios encountered in real-world scenarios. The testing mask is randomly generated based on the algorithm from (Li et al., 2022), with the histogram of the testing mask ratio of COCO dataset visualized in Fig. 6.

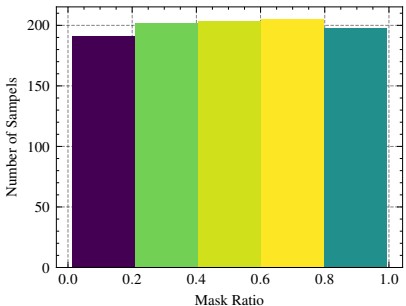

Figure 6: Illustration of mask ratio.

All quantitative experiments are conducted on the COCO (Lin et al., 2014) and Places (Zhou et al., 2018) datasets. Evaluation of the methods involves using the first 1000 images in the COCO validation set and the first 5000 images in the Places validation set. Masks from COCO are replicated five times for the Places dataset. For auxiliary guided completion, the Canny algorithm (Canny, 1986) and PiDiNet (Su et al., 2021) are employed to obtain the canny edge map and sketch map, respectively. MiDaS (Ranftl et al., 2022) is adopted to acquire depth maps for both datasets. COCO serves as a dataset with semantic segmentation and prompt text annotations. As the Places dataset lacks ground-truth labels, the semantic segmentation map is estimated using CIRKD (Yang et al., 2022).

## B  Application Results

### B.1  Real User-input Image Editing

We highlight the adaptability of our method in handling *user-input image editing* tasks designed to manipulate real-world images based on user intention, as demonstrated in Fig. 15. This figure emphasizes our method's capacity to modify the structure or semantics of local regions using user-input guidance such as scribble, pose map and prompt text, while fully maintaining the integrity of the unmasked region.

### B.2  Image Outpainting

Our method can also be used to extend an image, like generating a panorama from a small part of the image content. As demonstrated in Fig. 16 and Fig. 17, our method showcases its capability to *outpaint* a photograph or a painting guided by text and sketch map. Remarkably, our method exhibit the ability to generate suitable content that is harmonious even with the broader context of a panoramic image.

## C  More Experimental Results And Studies

### C.1  Quantitative Comparisons with Conditional Text-to-Image Methods

Contemporary methods like ControlNet and T2I-Adapter have demonstrated remarkable achievements in controllable image generation. For a direct comparison, we employ latent-level blending to utilize these methods for image completion, maintaining the experimental settings of earlier Experiments.

| | COCO | | Places2 | | |
|---|---|---|---|---|---|
| Method | FID↓ | PickScore↑ / % | FID↓ | U-IDS↑ / % | P-IDS↑ / % |
| ZITS ♠ | 61.27 | 28.09 | 18.96 | 18.75 | 7.20 |
| T2I-Adapter ♠ | 48.23 | 30.10 | 10.39 | 19.44 | 5.66 |
| ControlNet ♠ | **37.17** | **37.30** | 10.35 | 18.45 | 4.58 |
| Ours ♠ | 41.15 | 34.94 | **8.32** | **26.23** | **10.96** |
| T2I-Adapter ♥ | 50.92 | 30.22 | 18.10 | 14.91 | 4.56 |
| ControlNet ♥ | 46.13 | 32.52 | 15.96 | 14.46 | 3.18 |
| Ours ♥ | **39.43** | **37.12** | **9.09** | **25.34** | **10.64** |
| T2I-Adapter ♣ | 50.65 | 28.10 | 15.36 | 15.99 | 4.30 |
| ControlNet ♣ | 58.27 | 26.11 | 18.13 | 13.68 | 3.24 |
| Ours ♣ | **41.91** | **34.96** | **10.27** | **24.21** | **9.93** |
| T2I-Adapter ◇ | 39.08 | 34.26 | 14.27 | 14.76 | 3.30 |
| Ours ◇ | **37.65** | **49.57** | **8.98** | **25.30** | **10.90** |

Table 4: Quantitative comparisons with conditional image completion and text-to-image methods. ♠: ground truth edge map as guidance, ♥: estimated depth map as guidance, ♣: segmentation map as guidance, ◇: using segmentation, depth, canny, sketch, and text (on COCO) for guidance simultaneously.

| | COCO | | | | Places2 | | |
|---|---|---|---|---|---|---|---|
| Method | CLIP↑ / % | PSNR↑ | SSIM↑ | LPIPS↓ | PSNR↑ | SSIM↑ | LPIPS↓ |
| ZITS ♠ | 28.33 | 14.31 | 0.2767 | 0.5382 | **21.07** | **0.6888** | **0.2614** |
| T2I-Adapter ♠ | 28.59 | 18.26 | 0.6272 | 0.3409 | 18.34 | 0.6537 | 0.3208 |
| ControlNet ♠ | 28.97 | **19.22** | **0.6871** | **0.3183** | 18.59 | 0.6647 | 0.3220 |
| Ours ♠ | **29.37** | 18.13 | 0.6188 | 0.3467 | 19.00 | 0.6569 | 0.3111 |
| T2I-Adapter ♥ | 28.05 | 17.84 | 0.5894 | 0.3729 | 17.57 | 0.5765 | 0.3805 |
| ControlNet ♥ | 28.22 | **18.16** | **0.6275** | **0.3583** | 17.49 | 0.5967 | 0.3703 |
| Ours ♥ | **29.11** | 17.47 | 0.5960 | 0.3628 | **17.91** | **0.6109** | **0.3432** |
| T2I-Adapter ♣ | 28.10 | **17.55** | 0.5635 | 0.3830 | 17.33 | 0.5529 | 0.3923 |
| ControlNet ♣ | 26.48 | 16.98 | 0.5587 | 0.4023 | 17.22 | 0.5568 | 0.3948 |
| Ours ♣ | **28.87** | 17.01 | **0.5681** | **0.3799** | **17.44** | **0.5860** | **0.3591** |
| T2I-Adapter ◇ | 30.23 | **19.45** | **0.6748** | **0.3217** | **19.17** | **0.6626** | **0.3255** |
| Ours ◇ | **31.29** | 17.49 | 0.5921 | 0.3717 | 17.85 | 0.6085 | 0.3439 |

Table 5: Additional quantitative comparison results in terms of CLIP score and traditional reconstruction metrics. ♠: ground truth edge map as guidance, ♥: estimated depth map as guidance, ♣: segmentation map as guidance, ◇: using segmentation, depth, canny, sketch, and text (on COCO) for guidance simultaneously.

As Table 4 reveals, our MaGIC significantly surpasses baseline models in FID, U-IDS, P-IDS, and PickScore for most guidance types. In multi-modality guidance, we enhance T2I-Adapter with multi-adapter controlling Mou et al. (2023) (feature-level addition), resulting in T2I-Adapter◇. For the COCO dataset, we employ five modalities: *canny edge, depth, segmentation, sketch map, and text*. For the Places dataset, we utilize *canny edge, depth, segmentation, and sketch map*, as it lacks manually-crafted captions. Our MaGIC outperforms T2I-Adapter◇ by 44.68% in PickScore on COCO, and shows improvements of 37.07%, 71.40%, and 230.30% in FID, U-IDS, and P-IDS, respectively, on Places, as detailed in the last two rows of Table 4.

While traditional reconstruction metrics, such as PSNR, SSIM, and LPIPS, rely on pixel-wise similarity to the ground truth and tend to favor blurry outputs, as noted by Zhao et al. (2021) and Li et al. (2022), they are not optimal for quantitatively assessing image completion. Nevertheless, we include these traditional metrics for reference. Additionally, we provide the CLIP Score as an extra measure for a more comprehensive evaluation in the Table 5.

## C.2 Qualitative Comparisons in Multimodal Conditioning

As illustrated in Figure 9, we perform qualitative, side-by-side comparisons with T2I-Adapter⋄. For our MaGIC⋄, we produce five diverse results. Under the guidance of four modalities, MaGIC demonstrates strong controllability and high-fidelity outputs, aligning with our quantitative findings. In contrast, while T2I-Adapter⋄ effectively adheres the layout or shape to guidance, it fails to generate images of above-average quality with realistic details. This shortfall is attributed to the feature-level addition approach, leading to an out-of-distribution effect in the SD U-Net.

## C.3 Study in Feature-level Addition

Although we claim that FLA (feature-level addition) is a simple yet imperfect method for combining multiple modalities, and demonstrate the effectiveness of our CMB by comparative experiments, a comprehensive understanding of these two methods remains elusive. To this end, we opt to visualize the feature distributions stemming from T2I-Adapter with single-modality training and multi-modality utilization strategies, including FLA or our proposed CMB.

In specific, we choose the feature from the middle denoising step (*i.e.*, the 25th step of DDIM sampler) and output from U-Net encoder. The

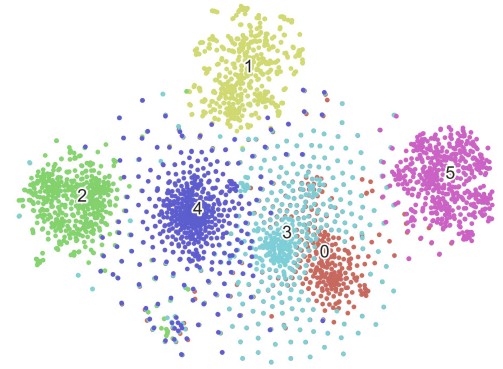

Figure 7: t-SNE visualization of features output from U-Net encoder.

t-SNE visualization result is shown in Figure 7, and different colors represents features from different sources, while the associated numbers indicate the index and cluster center of each feature type. Numbers ranging from small to large represent features obtained from T2I-Adapter-Canny (0), T2I-Adapter-Depth (1), T2I-Adapter-Segmentation (2), T2I-Adapter-Sketch (3), T2I-Adapter-CMB (4) and T2I-Adapter-FLA (5), respectively, where the first four indicate the trained single-modality while the last two are two methods of combining these four modalities.

We can draw two conclusions from Figure 7:

1. Features derived from different single-modality models (0, 1, 2 and 3) show significant distribution disparities, and FLA (5) directly adds modality features resulting in the distribution deviation of obtained feature from all others. This observation aligns with our assertion in the main manuscript that "we called feature-level addition is impractical, as the denoiser is trained solely on the distribution of $\hat{F}_c = F_{enc} + F_c$".

2. In contrast to FLA, the distribution of features obtained through CMB (4) is surrounded by other single-modality distributions. This phenomenon is coherent with Equation 4 and 5, where the distribution of obtained features is "pulled" by the distributions of the other four single-modalities.

## C.4 Failure Cases

Figure 8 shows two failed instances of applying CMB on T2I-Adapter for multi-modalities guidance. Here we present a more complex testing scenario involving non-overlapping information between two modalities. In the first case, we use Anything-4.0 as the backbone and there are two mistakes in the generated image: the misshapen cat and the incorrectly positioned bench under the girl. The former discrepancy possibly arises from the pose adapter contributing stronger features compared to those from the depth adapter, consequently affecting the representation of the latter information, which is not accurately reflected in the generated image. The issue might be alleviated by training depth adapter more stronger or increasing $\delta_{depth}$ while decreasing $\delta_{pose}$ (see Equation 4 for details). While the latter mistake is the inherent challenge in SD, and many related works Chen et al. (2023); Chefer et al. (2023) could be referenced for potential mitigation steategies. In the second example, the generated image depicts a wall as the background, leading to the complete loss of depth information from the depth map.

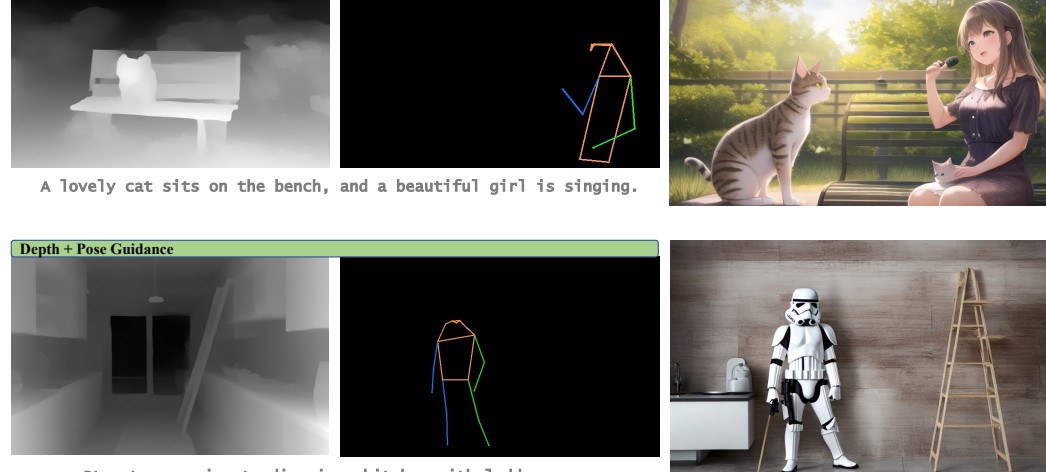

Figure 8: Failed cases when adopting CMB on T2I-Adapter.

These instances underscore that the inherent problems of SD persist despite employing CMB. Furthermore, in scenarios where information correlation between different modalities is low, evident issues such as information loss and errors in generated images become more pronounced. Through these failed cases, it can be seen that the inherent challenges of SD cannot be eliminated by CMB. Furthermore, in scenarios where information correlation between different modalities is low, evident issues such as information loss and errors in generated images become more pronounced.

## C.5 SELECTION OF $\delta_C$

In our quantitative and qualitative comparative experiments, it natural to assign a constant weight of 1 to each $\delta_c$. However, it also could be considered $\delta_c$ as a hyperparameter. Notably, although adopting $\delta$ introduces a potential hyperparameter requiring tuning, our approach, focused on training-free scenarios, makes this addition not overly burdensome compared to the process of training a large, multimodal guided model.

## C.6 STUDY IN ADAPTABILITY AND IMAGE GENERATION APPLICATION

Our proposed MaGIC has the ability to adapt to a variety of backbone diffusion models, including but not limited to, the image generation model Anything-4.0, Stable Diffusion-1.5 (also employed by T2I-adapter and ControlNet), and the image completion model Stable Diffusion Inpainting-2.1 (the default in MaGIC). In order to elucidate the differences among these backbone diffusion models, a qualitative experiment was carried out, focusing primarily on the anime-style image generation model Anything-4.0, image generation model Stable Diffusion (SD), the mask-aware T2I-adapter (Mou et al., 2023; Avrahami et al., 2023), and our own MaGIC.

As portrayed in Fig. 12(a) and (b), our MaGIC method exhibits exceptional generalizability to image generation backbones. These backbones can produce convincing results guided by factors such as sketch, depth, segmentation, and the canny edge map. T2I-Adapter (Mou et al., 2023) is a conditional image generation framework based on Stable Diffusion-1.5. To equip T2I-Adapter with CMB for the completion of a masked image, we implemented a technique known as latent-level blending (Avrahami et al., 2023). As evidenced in Fig. 12(c) and (d), incorporating blending into T2I-Adapter can preserve the unmasked region remains while the generated masked region does not perceive the unmasked region, given the fact that there are two sheep heads in a single sheep.

We further adapt CMB to T2I-Adapter and ControlNet (Zhang & Agrawala, 2023) to verify its effectiveness on multi-modality guided image generation. The results are shown in Fig. 13 and Fig. 14.

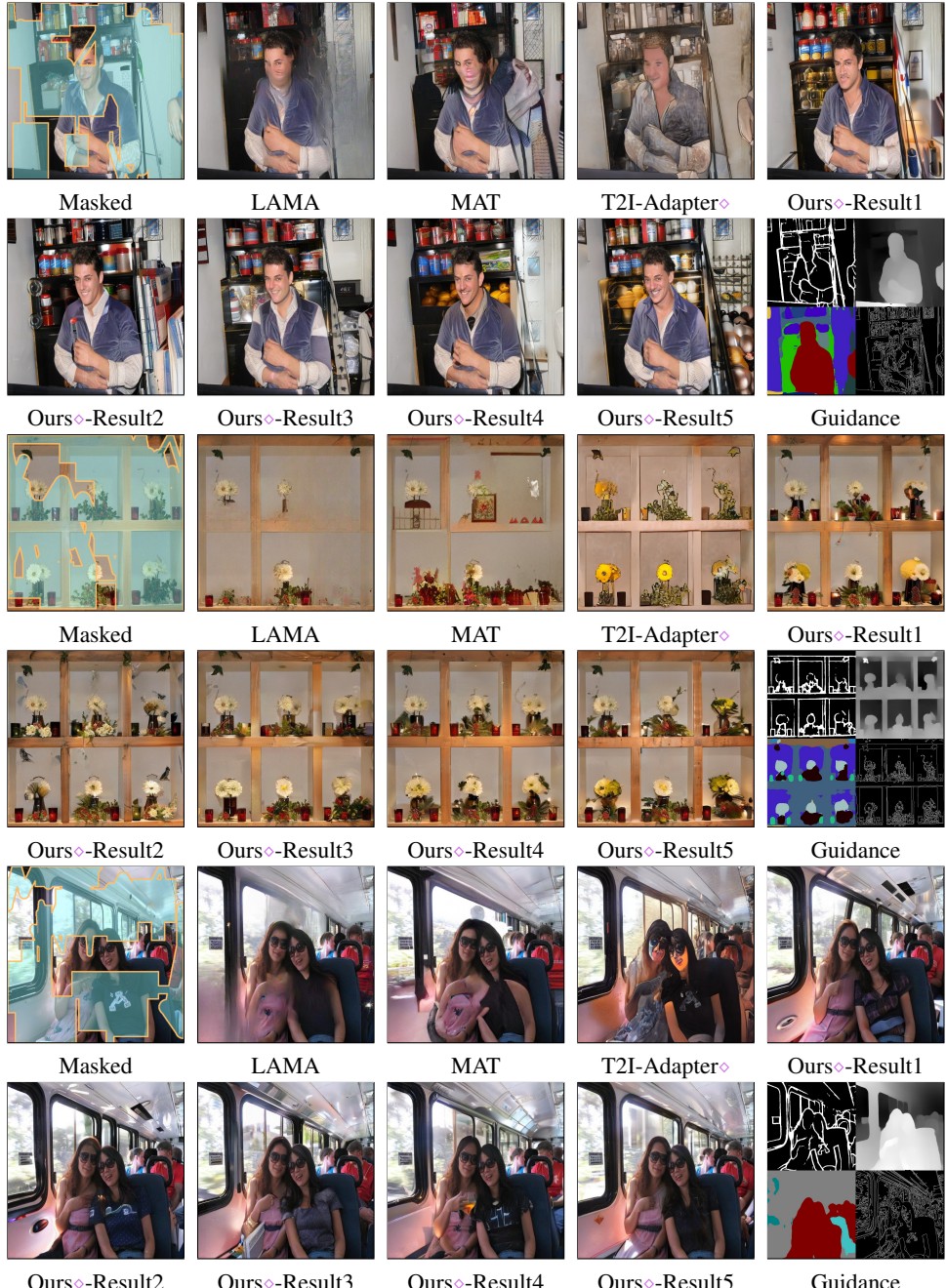

Figure 9: Qualitative results of MaGIC with four guidance compared to baselines.

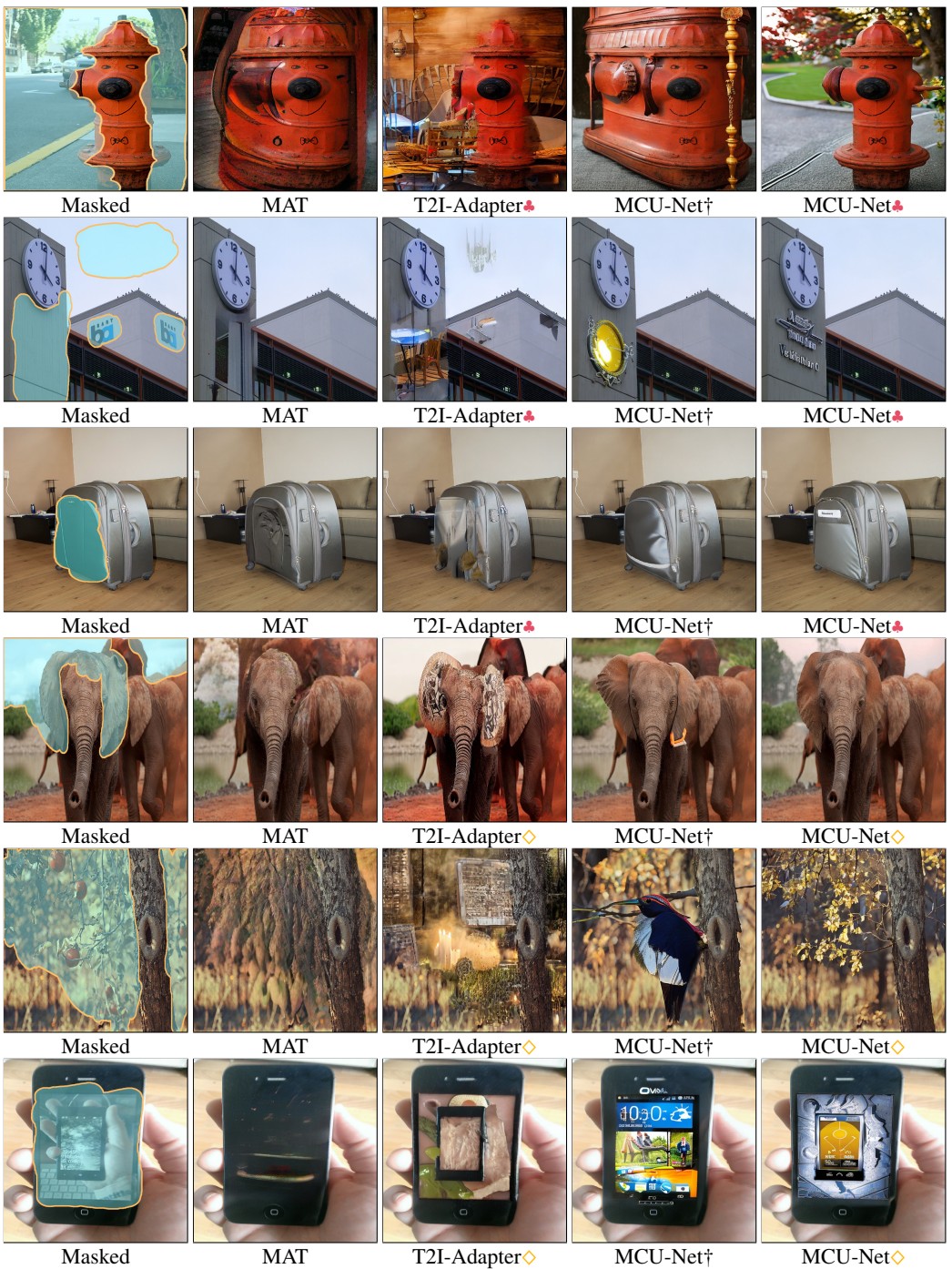

Figure 10: Qualitative results of MCU-Net compared to T2I-Adapter and MAT. ♣: segmentation map as guidance, ◇: sketch map as guidance. MCU-Net† denotes the completion without any guidance.

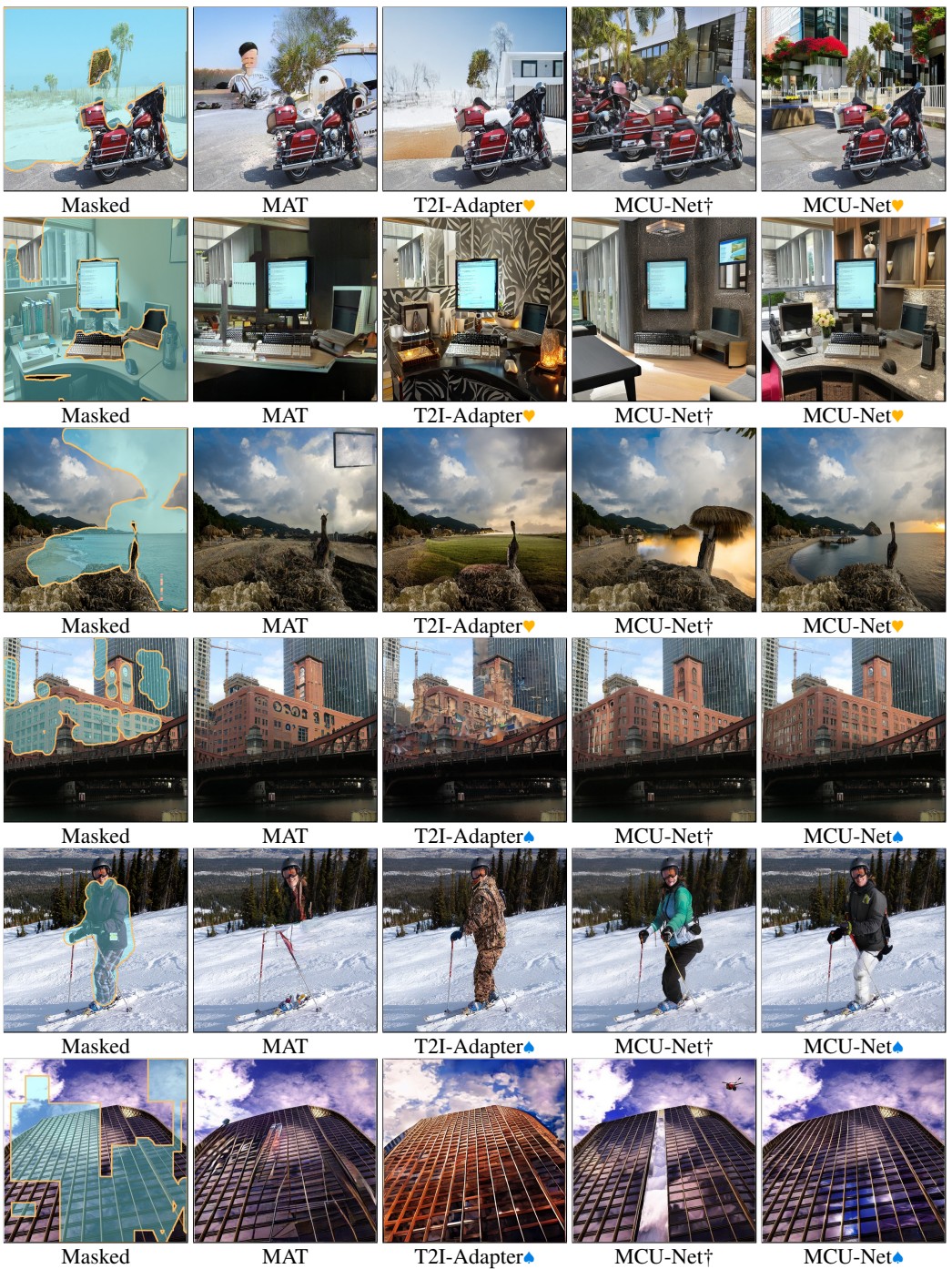

Figure 11: Qualitative results of MCU-Net compared to T2I-Adapter and MAT. ♥: depth map as guidance, ♠: canny edge as guidance. MCU-Net† denotes the completion without any guidance.

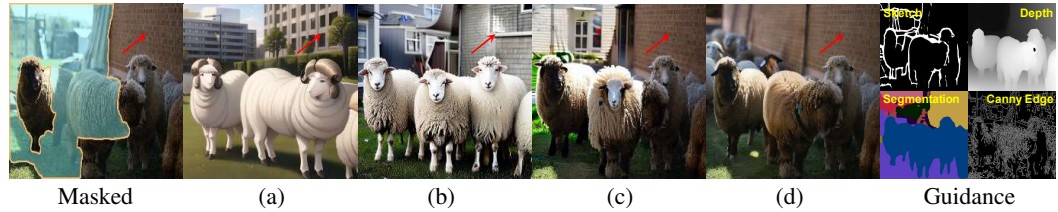

| Masked | (a) | (b) | (c) | (d) | Guidance |

Figure 12: Results from different backbone models. (a) Anything-4.0, (b) Stable Diffusion-1.5, (c) T2I-adapter with blending, (d) Stable Diffusion Inpainting-2.1 (default in MaGIC).

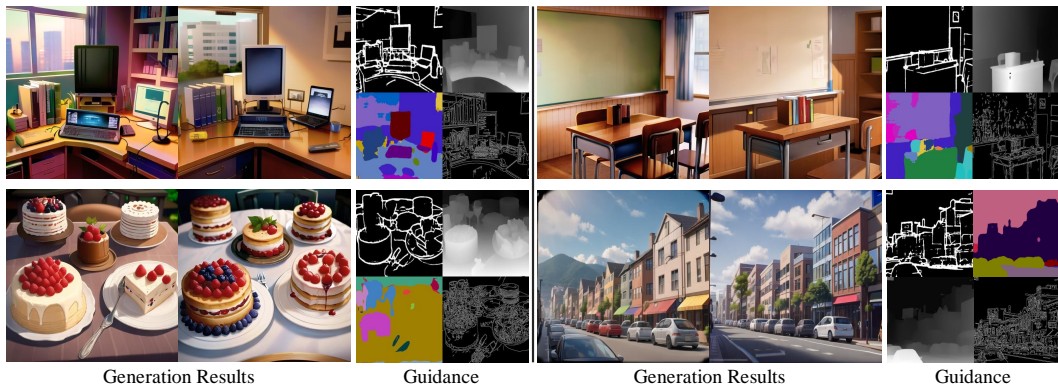

| Generation Results | Guidance | Generation Results | Guidance |

Figure 13: Adapting the proposed CMB to T2I-Adapter (Mou et al., 2023) with Anything-4.0 backbone for multi-modality guided image generation.

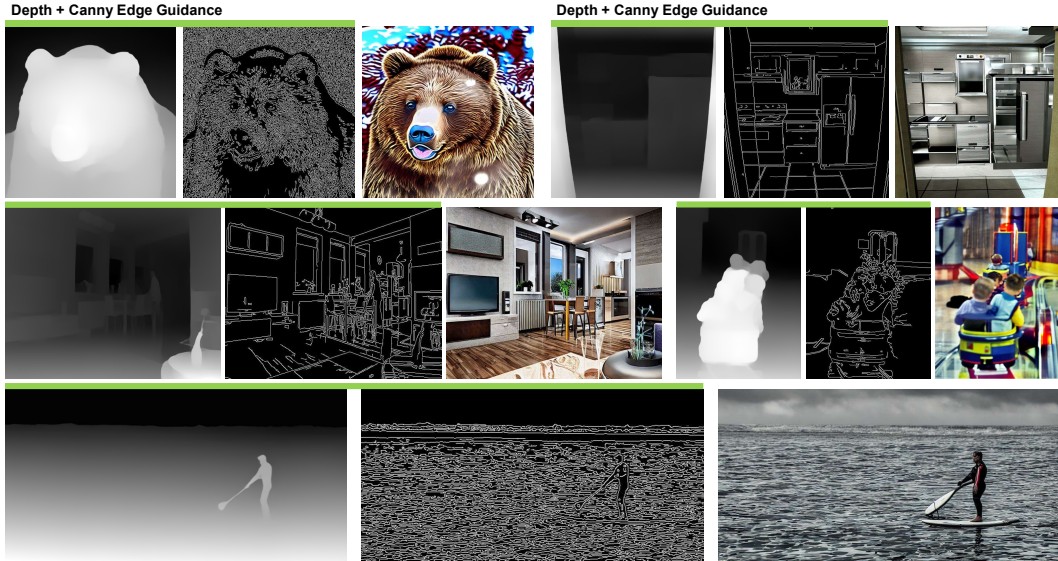

Figure 14: Adapting the proposed CMB to ControlNet (Zhang & Agrawala, 2023) for multi-modality guided image generation.

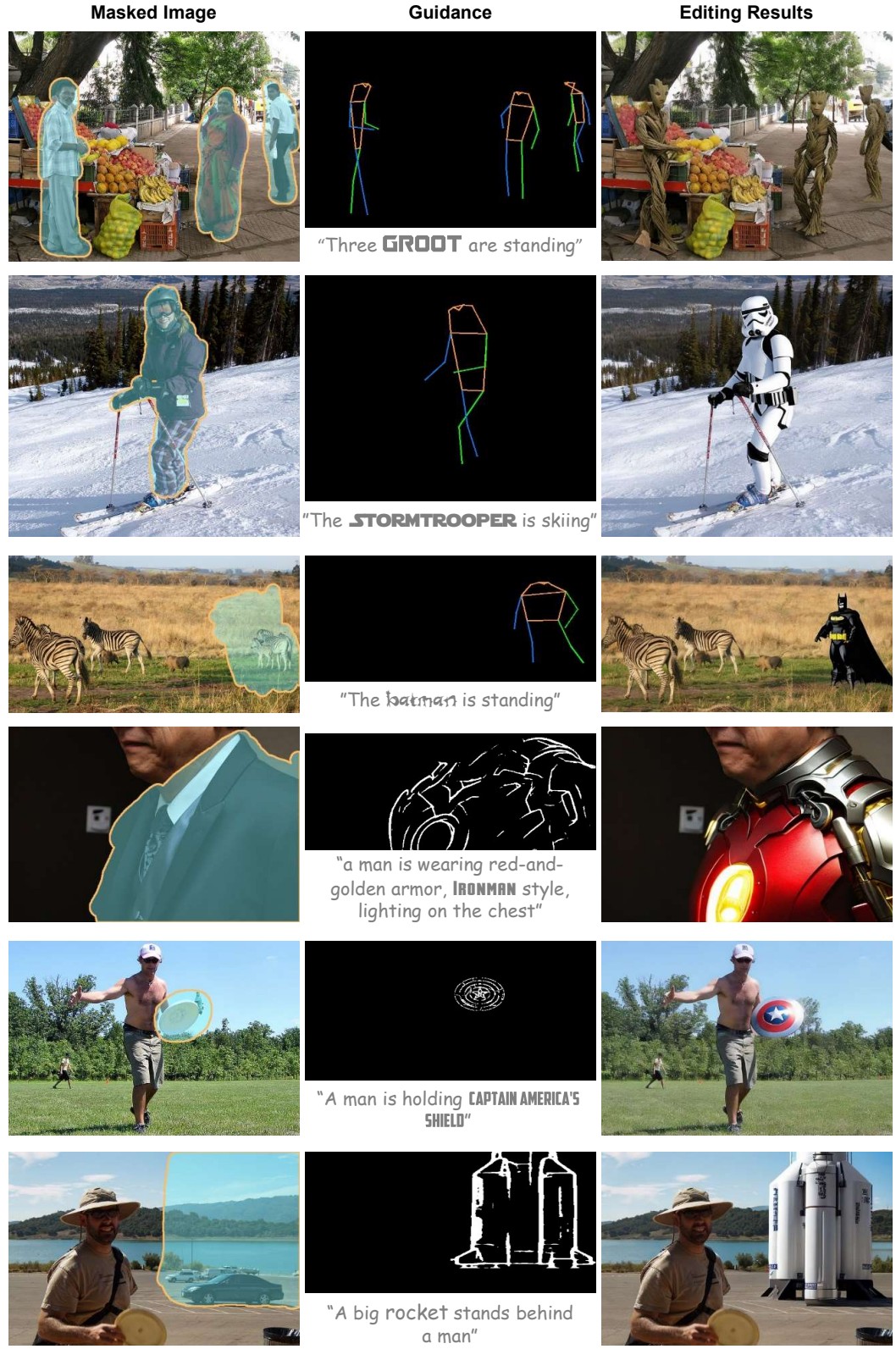

Figure 15: Application examples: local editing.

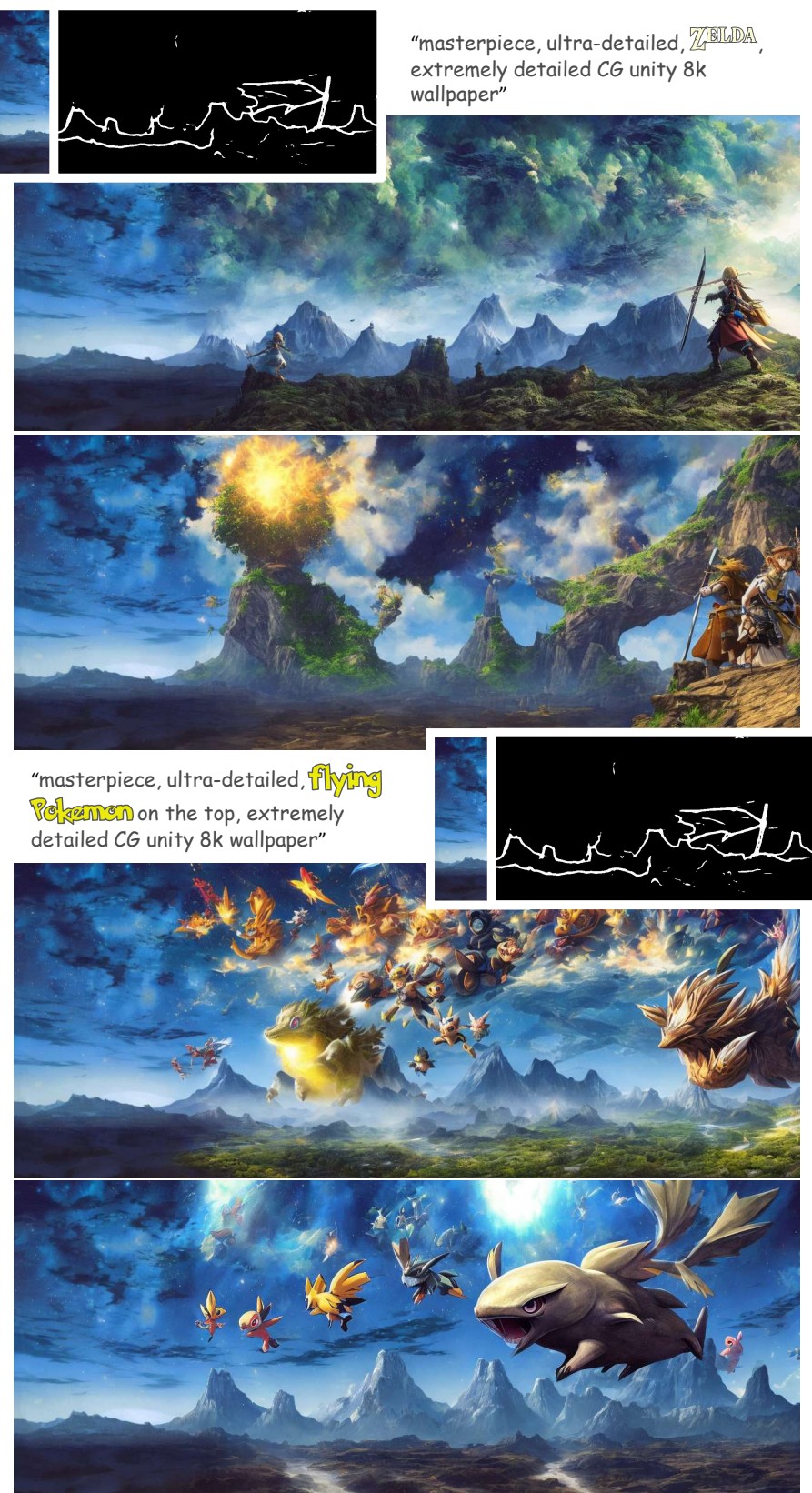

Figure 16: Application examples: sketch and text guided image outpainting.

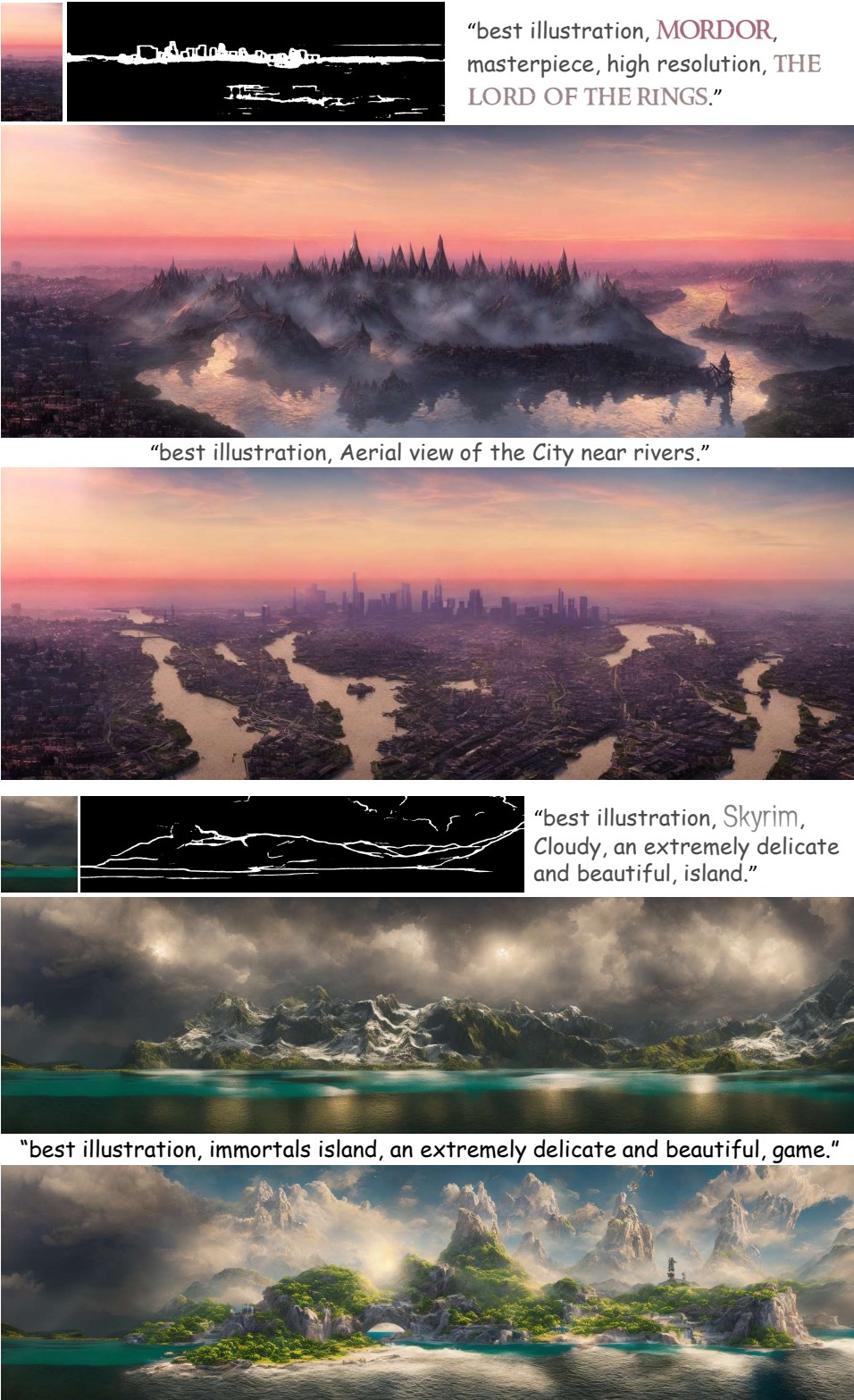

Figure 17: Application examples: sketch and text guided image outpainting.

