# OpenReview forum: "MaGIC: Multi-modality Guided Image Completion"
_ICLR.cc/2024/Conference — ICLR 2024 poster_

### Official Review · Reviewer_VxR9 · 2023-10-31

**Soundness:** 3 good
**Presentation:** 3 good
**Contribution:** 3 good
**Rating:** 8
**Confidence:** 4

**Summary:**

This paper proposes a multi-modal approach for image completion with LARGE missing regions. The different modalities, such as depth, edge, sketch, pose, provide complementary information for plausible completion. The approach does not require training.

**Strengths:**

1. Dealing with LARGE missing regions is a critical task in image completion. This topic is of broad interest in the ML and image processing community.
2. The idea of leveraging multiple resources is nice though not ground-breaking novel. Making is scalable and flexible is the key, which is solved by two stage approach: modality oriented conditional network and across-modality blending.
3. The approach is integrated into the diffusion process neatly and training-free.
3. The paper is very well written and easy to follow, with good illustrations.
4. The results are convincing with well-planned experiments, which also demonstrate good image generation results beyond completion

**Weaknesses:**

1. I'm not fully convinced that different image channels/features, such as depth, sketch, edge, could be called modality.
2. The fair comparison is not easy since most SOTA are not considering multiple resources in the same time. It'd be nice to share some insight into this, and share failure cases.

**Questions:**

As in weaknesses.

---

> ### Author Response · Authors · 2023-11-19
> **Rebuttal**
>
> Thank you for the time, thorough comments, and valuable suggestions. We are pleased that you acknowledged our novel idea, the very well-written paper, and our convincing experiments.
>
> > **Q1**: I'm not fully convinced that different image channels/features, such as depth, sketch, edge, could be called modality.
>
> **A1**: Sorry for any confusion caused. We fully understand that in certain fields, 'modality' usually refers to forms with greater differences, such as text, video, and audio, rather than sketch and canny edge. However, in our context, we adhere to the terminology and conventions established by preceding studies [1-3].
>
> `[1]` Toward multimodal image-to-image translation. NeurIPS. 2017.
>
> `[2]` Unbiased multi-modality guidance for image inpainting. ECCV. 2022.
>
> `[3]` Multimodal Conditional Image Synthesis with Product-of-Experts GANs. ECCV. 2022.
>
> > **Q2**: The fair comparison is not easy since most SOTA are not considering multiple resources in the same time. It'd be nice to share some insight into this, and share failure cases.
>
> **A2**: We deeply appreciate your observation regarding the difficulty in constructing a fair comparison, given the absence of baselines with similar multi-conditioning features. In response, we have included an analysis of failure cases in Figure 8 (Appendix) in our updated version.
>
> Please let us know if there are further questions. Thanks again!

---

> ### Author Response · Authors · 2023-11-22
> **A friendly reminder**
>
> Dear Reviewer,
>
> I would like to send a kind reminder. Has our response addressed your concerns? The reviewer discussion period is nearing its end, and we eagerly await your reply. Your suggestions and comments are invaluable to the community. Thank you!
>
> Best, The authors

---

### Official Review · Reviewer_jMgg · 2023-11-02

**Soundness:** 2 fair
**Presentation:** 2 fair
**Contribution:** 1 poor
**Rating:** 6
**Confidence:** 3

**Summary:**

MaGIC or Multi-modality Guided Image Completion can merge text, canny edge, sketch, segmentation, depth, pose, or any arbitrary combination as guidance for image completion. The authors aim to design a framework that is scalable and flexible. MaGIC has two core components -- a modality-specific U-Net (MCU-Net), and a consistency modality blending (CMB). The MCU-Net will be individually fine-tuned under each single modality, in the first stage. Then, to achieve multi-modality guidance, the CMB algorithm flexibly aggregates guidance signals from any combination of previously learned MCU-Nets. The MCU-Net is similar to T2I-Adapter, composed of a standard U-Net denoiser from the pre-trained Stable Diffusion (SD) and an encoding network which injects a single modality guidance into the U-Net to attain single-modality guidance. The CMB leverages guidance loss to gradually narrow the distances between intermediate features from SD pre-trained U-Net and multiple MCU-Nets during denoising sample stage. This ensures that the SD U-Net features do not deviate too much from the original feature distribution during multi-modality guidance. CMB is training-free and allows for the flexible addition or removal of guidance modalities, avoiding cumbersome re-training and preserving the feature distribution of the original SD U-Net denoiser.

To verify MaGIC, the authors conduct experiments on image inpainting, outpainting, and editing using the COCO, Places2, and in-the-wild data.

**Strengths:**

I like the extension of classifier guidance to multiple modalities that too training-free. Similar techniques has been explored in other single-modality context like in Sketch-Guided Text-to-Image Diffusion Models, but extending to multi-modal case is a nice extension.

The qualitative comparisons are very intuitive (especially with T2I-Adapter and ControlNet). The overall presentation is reasonable and easy to follow.

The authors included substantial appendix sections, detailing several architectural details like the design of MCU-Net.

**Weaknesses:**

While this is an interesting piece of work, I have some big gripes (please let me know if I understood it wrong):

In related work section (page 3 last paragraph), the authors give the impression that T2I-Adapter (and ControlNet) "fails to simultaneously use multi-modality as guidance". This is completely wrong. I understand that T2I-Adapter do not explicitly train jointly for multiple modalities, but they can combine multiple modalities (see section 4.3.2 in T2I-Adapter).

Second, the authors need to provide a solid justification why a simple "feature-level addition" mentioned in Page-6 paragraph-2 is not good. T2I-Adapter (in broad terms) does exactly that. I would need more detailed experiment and intuition comparing both T2I-Adapter and ControlNet for multi-modality case.

The argument given by authors "denoiser is trained solely on the distribution of $$\hat{F}_{c} = F_{enc} + F_{c}$$ " makes even less sense when you consider ControlNet with its zero-convolutions. In ControlNet we get plausible generation from the starting iterations, thanks to zero-convolutions and with training the conditioning branch becomes good. Hence, the distribution mismatch should not really be an issue to begin with.

Apart from my major concerns above, there are some minor corrections/concerns:
1. I think Eq. 4 should be $$ l (\hat{F}_{C}, F_{*}) = \frac{1}_{L} \sum_{l=0}^{L} \sum_{i=1}^{N} \delta_{c} || \hat{F}_{c}^{l} - F_{*}^{l} ||_{2}^{2} $$ (Note: subscript is C and not c?)

2. Typos e.g., 2nd last paragraph just after equation 2 "Denoising"

3. The MCU-Net is basically T2I-Adapter. I do not see any reason to have a new name for it (only to rebrand something and create more confusion). On the other hand, given MCU-Net is same as T2I-Adapter, the only merit of this paper is CMB.

**Questions:**

Since the only contribution of this paper is CMB, I would suggest to have a very detailed comparison with respect to T2I-Adapter, ControlNet, and many more (for multiple modalities).

Apart from just a few qualitative results and some incremental metrics improvement, why do you think Converse Amplification (or simply a variant of classifier guidance) a better approach than zero-convolutions with ControlNet?

Also, can you add some failure cases of CMB? This is important to give a better idea of where ControlNet lacks and where CMB lacks (I understand CMB can be coupled with ControlNet or T2I-Adapter).

---

> ### Author Response · Authors · 2023-11-19
> **Rebuttal (part 1/2)**
>
> Thank you for the time, thorough comments, and nice suggestions. We are pleased to clarify your questions step-by-step.
>
> > **Q1**: In related work section (page 3 last paragraph), the authors give the impression that T2I-Adapter (and ControlNet) "fails to simultaneously use multi-modality as guidance". This is completely wrong. I understand that T2I-Adapter do not explicitly train jointly for multiple modalities, but they can combine multiple modalities (see section 4.3.2 in T2I-Adapter). Second, the authors need to provide a solid justification why a simple "feature-level addition" mentioned in Page-6 paragraph-2 is not good. T2I-Adapter (in broad terms) does exactly that.
>
> **A1**: Thank you for your thorough review. We acknowledge that the T2I-Adapter represents an outstanding contribution to controllable image generation, exhibiting exponential performance in single-modality guidance tasks.
> However, our claim that it "**fails** to simultaneously use multi-modality as guidance" remains valid.
> The T2I-Adapter, as outlined in its paper, implements multimodal guidance through a straightforward feature-level addition approach, while showcasing results with bimodal guidance.
> However, theoretically, this feature-level addition leads to a distribution mismatch with the pre-trained model, a phenomenon illustrated in Figure 7 of our updated paper. This implies that with an increased number of guiding modalities (more than two), such straightforward strategy can drastically perturb the distribution, resulting in distorted outputs.
> Empirically, our hypothesis is corroborated in Appendix Table 4 and Figure 9 of our updated version. In Table 4, our approach shows improvements of 230.30%, 71.40%, and 37.07% in image quality metrics P-IDS, U-IDS, and FID, respectively, compared to the multimodal guidance of T2I-Adapter.
> Qualitatively, Figure 9 demonstrates that while the T2I-Adapter aligns well with the shape and layout of multiple guiding modalities, it produces images lacking in realism. The loss of high-frequency details and the failure to restore content in unmasked regions are due to the simultaneous addition of features from four modalities during inference, which does not match the distribution of the T2I-Adapter's backbone U-Net.
>
> > **Q2**: I would need more detailed experiment and intuition comparing both T2I-Adapter and ControlNet for multi-modality case.
>
> **A2**: Thanks for your suggestion. As highlighted by Reviewer VxR9, the absence of multimodal guided image completion methods makes it challenging to conduct fair comparative experiments. ControlNet and T2I-adapter are specialist in various single modality image generations, and we implement a latent-level blending to enable them to perform image completion. This way ensures maximum fairness in comparing the best multimodal guided image completion methods currently available.
> We have put the Table 4 (Appendix) for detailed quantitative experiments with T2I-Adapter and ControlNet in our new version. Additionally, in the Figure 9 (Appendix), we also provide qualitative comparison with T2I-Adapter under multi-modal conditioning. This supports our findings in Section 4.2, where T2I-adapter* and ControlNet* are observed to fall short in preserving spatial consistency.

---

> ### Author Response · Authors · 2023-11-19
> **Rebuttal (part 2/2)**
>
> > **Q3**: The argument given by authors "denoiser is trained solely on the distribution of $\hat{F_c} = F_{enc} + F_c$" makes even less sense when you consider ControlNet with its zero-convolutions. In ControlNet we get plausible generation from the starting iterations, thanks to zero-convolutions and with training the conditioning branch becomes good. Hence, the distribution mismatch should not really be an issue to begin with.
>
> **A3**: Thanks for raising such an interesting question,  and we would like to provide a comprehensive answer addressing two crucial aspects.
>
> 1. **The function of zero-convolution:** Our understanding of zero-convolution aligns with its purpose in ControlNet for stabilizing the initial training process. This technique avoids introducing "noise" to deep features generated by production-ready weights, as discussed in ControlNet. In both T2I-Adapter and ControlNet, the modality features are directly added to the features of the original Stable Diffusion, which is trained for conditional generation, and that is why we said the denoiser is trained on $\hat{F_c} = F_{enc} + F_c$. Consequently, we disagree with there exists any direct correlation between the zero-convolution and training distribution.
> 2. **The origin of distribution mismatch:** Our primary focus resolves around the process of combining separately trained condition modules (such as adapter in T2I-Adapter) to support multi-modality input in a training-free manner. Analogous to how modality features might act as noise during the initial training phase for a well-established Stable Diffusion, we posit that features produced by separately trained conditional modules could similarly be perceived as noise by each other. For example, suppose the trained networks supports segmentation ($\hat{F_{seg}} = F_{enc} + F_{seg}$) and depth $\hat{F_{dep}} = F_{enc} + F_{dep}$ modalities individually, the combined features $F_{seg}+F_{dep}$ could be noise inputs for both networks, leading to distribution mismatch.
>    We apologize for any confusion in our initial manuscript concerning this question and welcome further discussion to clarify these aspects.
>
>
> > **Q4.1**: I think Eq. 4 should be $\ell(\hat{F_C}, F_*) = \frac{1}{L} \sum_{l=0}^{L} \sum_{i=1}^{N} \delta_{c} || \hat{F_c^l} - F_*^{l} ||_{2}^{2}$ (Note: subscript is C and not c?)
>
> **A4.1**: Thanks for your nice suggestion. We have revised the equation 4 as:
>
> $\ell(\hat{F_\mathcal{C}}, F_*) = \frac{1}{L} \sum_{l=0}^L \sum_{c\in \mathcal{C}} \delta_{c} \| \hat{F_c^l} - F_{*}^l\|_2^2$
>
> > **Q4.2**: Typos e.g., 2nd last paragraph just after equation 2 "Denoising"
>
> **A4.2**: Thanks for your detailed review. We have corrected the typo in our new version.
>
> >**Q4.3**: The MCU-Net is basically T2I-Adapter. I do not see any reason to have a new name for it (only to rebrand something and create more confusion). On the other hand, given MCU-Net is same as T2I-Adapter, the only merit of this paper is CMB.
>
> **A4.3**: Sorry for confusion. It's important to clarify that MCU-Net is distinct from the T2I-Adapter; we cannot simply use the T2I-Adapter as our backbone for single modality guided image completion. Our work involves gathering multi-modality datasets and training MCU-Net with masked images. This training enables our model to adeptly fill masked regions while ensuring harmony with the unmasked areas. This way significantly contributes to our method's superior performance over baseline models, as evidenced by our experimental results.
>
> > **Q5**: I would suggest to have a very detailed comparison with respect to T2I-Adapter, ControlNet, and many more (for multiple modalities).
>
> **A5**: Thanks. We have included both quantitative and qualitative comparisons with respect to T2I-adapter and ControlNet. Please refer to Table 4 and Figure 9 in our updated version.
>
> > **Q6**: Apart from just a few qualitative results and some incremental metrics improvement, why do you think Converse Amplification (or simply a variant of classifier guidance) a better approach than zero-convolutions with ControlNet?
>
> **A6**: Thank you. As **A3** clarified, zero-convolution differs significantly from Converse Amplification in both motivation and implementation. If there are any further questions or concerns, please feel free to discuss them.
>
> > **Q7**: Also, can you add some failure cases of CMB? This is important to give a better idea of where ControlNet lacks and where CMB lacks (I understand CMB can be coupled with ControlNet or T2I-Adapter).
>
> **A7**: Thank you for your thoughtful suggestion. We have included the failure cases in Figure 8 in our revised version.
>
> Please let us know if there are further questions. Again, thanks!

---

> ### Author Response · Authors · 2023-11-22
> **A friendly reminder**
>
> Dear Reviewer,
>
> I would like to send a kind reminder. Has our response addressed your concerns? The reviewer discussion period is nearing its end, and we eagerly await your reply. Your suggestions and comments are invaluable to the community. Thank you!
>
> Best, The authors

---

> ### Author Response · Authors · 2023-11-22
> **Update on comparing T2I-Adapter with MCU-Net**
>
> Thank you again for your thoughtful review and comments. Following our discussion with Reviewer c8RF, we have further updated the qualitative experiments comparing T2I-Adapter with MCU-Net in the latest version (see Appendix A.3, Figures 10 and 11). This additional experiment supplements Figure 5, demonstrating that the T2I-adapter lacks awareness of the unmasked region, hence failing to maintain spatial consistency during image completion – a key contribution of our MCU-Net. Additionally, we'd like to gently point out that the main emphasis of our paper is centered on train-free multimodal guidance. We hope this clarification on the novelty of our work addresses your concerns and averts any potential reason for rejection.
>
> If your issues have been satisfactorily resolved, we kindly request you to consider not rating our work negatively. We deeply value your feedback and thank you for your time and attention.

---

> ### Author Response · Authors · 2023-11-23
> **Official Comment by Authors**
>
> As the phase of author-reviewer discussions draws to a close, we are so pleased to note your recognition of our efforts and the raised scores.
>
> We are grateful for the valuable suggestions you posed and appreciate the time and effort you devoted to the review process.

---

### Official Review · Reviewer_c8RF · 2023-11-07

**Soundness:** 2 fair
**Presentation:** 2 fair
**Contribution:** 3 good
**Rating:** 6
**Confidence:** 4

**Summary:**

The paper MaGIC: Multi-modality Guided Image Completion introduces a novel framework for image completion that supports various guidance modalities, such as text, edge, sketch, and more. The proposed method, MaGIC, enables flexible and scalable multi-modality guidance without the need for retraining the model. The paper demonstrates consistent improvements in image quality over existing approaches.

**Strengths:**

**Innovative and Flexible Approach** The paper addresses the challenging problem of multi-modality-guided image completion. It proposes a new simple training-free procedure, allowing for various guidance modalities, such as text, edge, sketch, segmentation, depth, and pose.

** Large Consistent Gains** The paper shows consistent and significant improvements over state-of-the-art approaches, particularly in image quality.

**Weaknesses:**

**Clarity and Typos** The paper is challenging to follow and contains multiple typos, which can impede understanding. Improved clarity in the presentation and thorough proofreading would enhance the paper's quality.

**Non-standard Update Scheme** The update scheme presented in equation (5) appears inhomogeneous, as it involves gradient descent with respect to $z_t$ but updates $z'_{t-1}$. This choice could be a reasonable heuristic but is not discussed or justified, which leaves questions about its validity.

**Lack of Quantitative Evaluation** The paper only qualitative results without quantitative evaluation metrics when compared with recent baselines such as ControlNet and T2I-adapter. In particular, the performances with respect to ControlNet should be carefully assessed.

**Inadequate Dataset and Modality Descriptions** The datasets used and the conditioning modalities are briefly presented. A more detailed description of the datasets, along with the rationale for their selection, would be beneficial.

**Missing Ablations** A more in-depth exploration of the impact of the weights $\delta_c$ in equation (4) would offer valuable insights. The stability of these parameters is critical as their tuning could rapidly be cumbersome.

**Inconsistent Results** The results in Table 3.b appear to be inconsistent, with FID scores not following the expected pattern. This raises questions about the efficiency of the CMB method and its need for complex hyperparameter tuning. In particular, for a fixed P=30, FID(Q=1)>FID(Q=10)>FID(Q=5).

**Questions:**

I wonder why the authors did not provide CLIP score evaluation as well as reconstruction performances.

---

> ### Author Response · Authors · 2023-11-19
> **Rebuttal (part 1/2)**
>
> Thanks for your constructive suggestions. Your endorsement of our method and experiments gives us significant encouragement. Here are our clarifications.
>
> > **Q1**: The paper is challenging to follow and contains multiple typos, which can impede understanding. Improved clarity in the presentation and thorough proofreading would enhance the paper's quality.
>
> **A1**: We are sorry for this and truly appreciate your feedback on improving our paper's clarity and addressing the typos. We have corrected the expression and typos in our new version.
>
> > **Q2**: The update scheme presented in equation (5) appears inhomogeneous.
>
> **A2**: Our approach aligns with what classifier-free guidance (CFG) [1] does and it makes common sense in this field. The core idea is to manually design reasonable loss and impose its gradient on latent via backpropagation. This process allows for the adjustment of the latent space towards generating the desired output. Please refer to applications [2-4] of CFG if you need further clarification. If there are any further questions or confusion, we are open to discussing them.
>
> `[1]` Classifier-Free Diffusion Guidance. Arxiv. 2022.
>
> `[2]` Freedom: Training-free energy-guided conditional diffusion model. Arxiv. 2023.
>
> `[3]` Training-free layout control with cross-attention guidance. Arxiv. 2023.
>
> `[4]` Unite and Conquer: Plug & Play Multi-Modal Synthesis using Diffusion Models. CVPR. 2023.
>
> > **Q3**: The paper only contains qualitative results without quantitative evaluation metrics when compared with recent baselines such as ControlNet and T2I-adapter.
>
> **A3**: Thanks for your valuable suggestion. We have added the Table 4 (Appendix) for quantitative evaluation with ControlNet and T2I-Adapter in our updated manuscript. Additionally, in the Figure 9 (Appendix), we also provide qualitative comparison with T2I-Adapter under multi-modal conditioning.
>
> >  **Q4**: The datasets used and the conditioning modalities are briefly presented. A more detailed description of the datasets, along with the rationale for their selection, would be beneficial.
>
> **A4**: We are sorry for the confusion of the datasets. But we actually have included a detailed description of the datasets for each modality in Appendix Section A.1 due to the space limitations.
>
> We chose the COCO dataset for its inclusion of manually-crafted segmentation. However, due to its relatively smaller image count compared to LAION, we opted for LAION to facilitate image filter. For instance, we retained only those images containing body to acquire pose maps, thereby preventing instances of empty guidance in the paired dataset. Our data selection and the underlying rationale align with those of ControlNet and T2I-Adapter, whose methodologies can be referenced for further information.
>
> > **Q5**: A more in-depth exploration of the impact of the weights $\delta_c$ in equation (4) would offer valuable insights. The stability of these parameters is critical as their tuning could rapidly be cumbersome.
>
> **A5**: Thank you for your feedback. As suggested, we have conducted ablation study on $\delta_c$ and the results are shown in the table below:
>
> | $\delta_c$ | FID$\downarrow$ | PickScore$\uparrow$/% |
> | :----------- | :---------------- | :---------------------- |
> | 1.0          | 37.65±0.22        | 49.57±0.17              |
> | 0.1          | 37.75±0.25        | 50.98±0.37              |
> | 0.5          | 37.27±0.26        | 50.96±0.15              |
> | 2.0          | 37.93±0.18        | 49.18±0.16              |
>
> In our initial manuscript, due to resource constraints and setting $\delta_c$ to 1 is trivial, we had not conducted this part of the experiment. However, the results suggest that adjusting the value within a practical range (0-1) yields improved results. This finding holds promise and offers valuable insights for future practical applications. Thanks again for the valuable suggestion.

---

> ### Author Response · Authors · 2023-11-19
> **Rebuttal (part 2/2)**
>
> > **Q6**: The results in Table 3.b appear to be inconsistent, with FID scores not following the expected pattern. This raises questions about the efficiency of the CMB method and its need for complex hyperparameter tuning. In particular, for a fixed P=30, FID(Q=1)>FID(Q=10)>FID(Q=5)."
>
> **A6**: Thank you for your detailed review. When Q is sufficiently large, the latent will match the conditioning modality. However, in the case of multimodal conditioning, a large Q can lead to stiff competition among multiple modalities for the latent. This competition can naturally lead to worse performance, as seen in metrics like FID. To clarify, we've emphasized in the revision that Table (b) is evaluated under multimodal conditioning.
>
> And we acknowledge the reviewer's concern regarding hyperparamter tuning. But our work targets training-free scenarios with introducing merely two additional hyperparameter. It is not excessively burdensome compared to training a large, multimodal guided model.
>
> > **Q7**: I wonder why the authors did not provide CLIP score evaluation as well as reconstruction performances.
>
> **A7**: PickScore, a **CLIP-based** metric, aligns more closely with human judgments, which is why we chose it. FID, a **reconstruction** metric, together with U-IDS and P-IDS, evaluates image quality and aligns well with human perception, making it suitable for our purposes. PSNR and SSIM, being **pixel-wise** metrics, are not ideal for **editing** or large mask settings [1,2]. Nevertheless, based on your suggestion, we have included Table 5 in the Appendix for a quantitative evaluation with ControlNet and T2I-Adapter in terms of CLIP score, and additional reconstruction metrics PSNR, SSIM, and LPIPS.
>
> `[1]` Large scale image completion via co-modulated generative adversarial networks. ICLR. 2021.
>
> `[2]` Mat: Mask-aware transformer for large hole image inpainting. CVPR. 2022.
>
> Please let us know if there are further questions. Thanks again!

---

> ### Author Response · Authors · 2023-11-22
> **A friendly reminder**
>
> Dear Reviewer,
>
> I would like to send a kind reminder. Has our response addressed your concerns? The reviewer discussion period is nearing its end, and we eagerly await your reply. Your suggestions and comments are invaluable to the community. Thank you!
>
> Best, The authors

---

> > ### Comment · Reviewer_c8RF · 2023-11-22
> >
> > Dear Authors,
> >
> > Thank you for providing a comprehensive rebuttal. It has clarified some of my concerns. Here are my responses to specific points:
> >
> > 1. **Concerning the guidance loss:** The choice to have different variables for the input and the output of the denoiser confused me. It appears clearer to me now. I believe this explanation could be beneficial to include in the paper.
> >
> > 2. **Concerning the evaluations:** I salute the effort for bringing in additional experiments. The performance gains observed seem convincing.
> >
> > 3. **Concerning the tuning of $\delta_c$:** I acknowledge the clarification about the constant choice of $\delta_c$ for all c. However, I would recommend explicitly mentioning this choice in the paper. The extra experiments tend to show that the performances are stable with respect to this parameter but it is difficult to really conclude on this point as the variation of $\delta_c$ was performed on a small range and with constant values for all weights.
> >
> > 4. **Question on the Novelty:** I tend to agree with Reviewer jMgg that the MCU-Net appears very similar to T2I-adapter. However, if I understand correctly, the MCU-Nets are trained as denoisers? A more explicit discussion on this point could be beneficial. In particular, could the authors be more specific on the exact training loss used for the MCU-Net? Would it be correct to say that the proposed method, in a nutshell, is given by multiple classifiers guidance with, instead of classifiers, modality-specific denoisers? This is not a good or a bad thing, I just want to be sure of my understanding.

---

> > > ### Author Response · Authors · 2023-11-22
> > >
> > > Thank you for your valuable feedback. Your affirmation and the discussion greatly benefit our work and the community atmosphere.
> > >
> > > We are pleased that our response has clarified your concerns, and that the additional experiments we conducted have convincingly addressed your questions regarding evaluations. Following your suggestion, we have now included an explanation of the guidance loss and clarified the selection of $\delta_c$ (Section 3.3 and Appendix C.5, respectively) in our latest revision. This ensures clear communication of these concepts to our readers. Here is a further clarification of MCU-Net:
> > >
> > > - Regarding the MCU-Net:
> > >
> > > You are correct in understanding that MCU-Net is trained as a single-modality conditional denoiser, using the classifier-free guidance loss $
> > > \arg\min_{\theta_c} \mathbb{E}\_{z_0,t,\epsilon \sim \mathcal{N}(0,I)} \\|\epsilon - \epsilon_\theta^t(z_t, m_\downarrow, x_{m\downarrow}, c) \\|_2^2.
> > > $
> > > It is also correct that MCU-Net is not a classifier but a modality-specific denoiser. We hope this further clarifies your understanding and invite more discussion if you have any questions.
> > >
> > > Regarding your issue about novelty, further clarification is necessary.
> > > T2I-Adapter implements conditional control via skip-connections in a straightforward manner, representing a significant contribution to controllable image generation. As mentioned in Section A.3 (appendix) of our original manuscript, we utilized the design of encoding network of the T2I-adapter, as our main focus is on train-free multimodal guidance.
> > > Although maintaining the same way of integrating modality features, our target is completely different from image generation handled by T2I-Adapter. In specific, masked images is used as input to our model instead of clear images, which means we have to retrain all the involved modalities separately instead of directly utilizing the pretrained models provided by T2I-Adapter. And the capability of our model is to proficiently fill in masked regions while ensuring spatial consistency with unmasked regions, which is proved to be features that T2I-Adapter does not possess (Figure 5).
> > > Beyond just textual descriptions, We have included qualitative comparisons between MCU-Net (without CMB) and T2I-Adapter in the latest version of our manuscript (Appendix A.3, Figures 10 and 11), highlighting their differences.
> > >
> > > We sincerely hope this clarification on the novelty addresses your concerns and prevents them from becoming a potential reason for rejection. If we have satisfactorily addressed your issues, we kindly remind you to avoid rating this work negatively. Your feedback is greatly appreciated. Thank you again!

---

> ### Comment · Reviewer_c8RF · 2023-11-23
>
> Thank you for the previous answer which helped me understand your method.
>
> In Fig. 10 and 11, are the completions given by the modality-specific MCU-Net or the denoiser $\theta^*$ guided by the MCU-Net?
>
> If I understand correctly, the performance gains brought by MAGIC come with the burden of training multiple modality-specific diffusion models which may require a large amount of data as well as high compute capacities while T2I-adapter is designed to be light-weight as it only attends to biased the encoding features. From these considerations, I have the feeling that the experimental set-up described in A.1. only describes the CMB sampling. Some insights on this point would be beneficial and could help understand if there are specific regimes where T2I-Adapter, ControlNet, or MAGIC  are expected to perform better.

---

> > ### Comment · Reviewer_c8RF · 2023-11-23
> >
> > Overall I am satisfied with the authors' rebuttal and I am thus willing to increase my raise.

---

> > > ### Author Response · Authors · 2023-11-23
> > > **Official Comment by Authors**
> > >
> > > Thank you sincerely for your engaged and insightful comments!
> > >
> > > Your constructive feedback has been instrumental in enhancing the clarity and contribution of our paper. The positive discussions we've had and the recognition of our efforts truly encapsulate the essence of the OpenReview process.
> > >
> > > Your time and efforts are immensely appreciated.

---

> > ### Author Response · Authors · 2023-11-23
> > **Official Comment by Authors**
> >
> > We sincerely appreciate your invaluable feedback and the opportunity to address your queries regarding our approach.
> >
> > Regarding Figures 10 and 11, please pay attention to the symbols and illustrations we provided, for example, MCU-Net with red plum flower symbol means the results derived from MCU-Net trained on segmentation maps.
> >
> > It seems that there are still some questions about MCU-Net training process and the proposed CMB strategy. To address these concerns comprehensively, let us offer a concise reintroduction:
> >
> > 1.  To augment the image completion task by incorporating additional cues to tackle the challenge of handling large missing regions or meeting user-specific completion expectation, we borrow the idea of integrating modality features to well-trained Stable Diffusion framework, inspired by T2I-Adapter and ControlNet. We trained several MCU-Net, utilizing Stable Diffusion Inpainting-2.1 as backbone (as elaborated in our supplementary), on different modalities such as segmentation maps, depth maps, keypose maps and so on.  And their effectiveness on single-modality image completion is fully proved by both qualitative (Figures 5, 10 and 11) and quantitative results (Table 1) in our manuscript. **It should be acknowledged that the backbone is frozen while only the modality encoding network ($\tau_{c^i}$ in our main text) is trained.**
> > 2. After obtaining MCU-Net that supports single-modality guidance, a practical question naturally arises: can these model extend support multi-modality guidance? T2I-Adapter proposed a straightforward solution: inputting multiple modalities into associated trained modality encoding networks and integrating them into the backbone. However, this approach yielded suboptimal results,  as elucidated in Figure 7 through t-SNE visualization on feature distributions. This led to the introduction of Co-Adapter by T2I-Adapter, which required retraining all modalities, incurring significant training costs. In contrast, we proposed CMB, which is a novel approach that utilizing the classifier-guidance to impose the multi-modality guidance on the original backbone features without extra training (which is appreciated by Reviewer `jMgg` and `VxR9`).
> >
> > So back to your questions, the efficiency of MCU-Net training is comparable to that of T2I-Adapter. Given our training strategies, we provided detailed information about the dataset training for each modality. Considering our equipment conditions (8 NVIDIA A100-40G), we conducted three-day training process for each modality, comprising approximately 100,000 iterations.
> >
> > Thanks again for your help on improving the quality of our work, and we will add the training details in our updated version of manuscript.

---

### Author Response · Authors · 2023-11-20
**General Response to Reviewers and ACs**

We sincerely thank the reviewers for their detailed and valuable comments.

In this post:
- (1) We summarize positive feedbacks from the reviews.
- (2) We summarize the changes to the updated manuscript.

### (1) Positive feedbacks

- **Strong empirical performance of the proposed method**
    - `[c8RF]`: *"The paper shows consistent and significant improvements over state-of-the-art approaches, particularly in image quality."*, *"The paper demonstrates consistent improvements in image quality over existing approaches"*
    - `[jMgg]`: *"The qualitative comparisons are very intuitive (especially with T2I-Adapter and ControlNet)."*
    - `[VxR9]`: *"...which also demonstrate good image generation results beyond completion"*
- **The proposed method**
    - `[c8RF]`: *"The paper addresses the challenging problem...It proposes a new simple training-free procedure"*, *"Innovative and flexible approach"*
    - `[jMgg]`: *"I like the extension of classifier guidance to multiple modalities that too training-free...extending to multi-modal case is a nice extension."*
    - `[VxR9]`: *"The idea of leveraging multiple resources is nice..."*, *"The approach is integrated into the diffusion process neatly and training-free."*
- **Experimental setup**
    - `[jMgg]`: *"The qualitative comparisons are very intuitive (especially with T2I-Adapter and ControlNet)."*, *"The authors included substantial appendix sections, detailing several architectural details..."*
    - `[VxR9]`: *"The results are convincing with well-planned experiments..."*
- **Presentation**
    - `[jMgg]`: *"The overall presentation is reasonable and easy to follow."*
    - `[VxR9]`: *"The paper is very well written and easy to follow, with good illustrations."*

### (2) Changes to the PDF

**Main text**

Minor fixes and wording improvements:

- `[c8RF,jMgg]` Proofreading (Fix typos) throughout paper, such as "availability" in abstract, "Denoising" in Section 3.1.
- `[jMgg]` Section 3.3, a fix to notation.
- `[c8RF]` Section 4.4, emphasize Table 3.b is evaluated on multimodal guidance.
- `[c8RF]` Section 3.3, an explanation of the guidance loss.

**Appendix**

- `[c8RF,jMgg]` Table 4 in Section C.1, an extensive quantitative comparison with T2I-Adapter and ControlNet.
- `[c8RF,jMgg]` Figure 9 and Section C.2, a qualitative comparison with T2I-Adapter under multimodal guidance.
- `[c8RF]` Table 5 in Section C.1, an additional quantitative comparison in terms of CLIP and reconstruction metrics.
- `[jMgg]` Figure 7 and Section C.3, a study in feature-level addition via t-SNE visualization.
- `[jMgg,VxR9]` Figure 8 and Section C.4, several failure cases.
- `[c8RF]` Section C.5, a clarification of selection of $\delta_\mathcal{C}$.
- `[c8RF,jMgg]` Figures 10 and 11 in Section A.3 highlight the differences in effectiveness between T2I-Adapter and MCU-Net.

We sincerely hope this work can bring some insights into the field of image completion. Thanks again to all reviewers for their valuable time to help improve our work. We remain enthusiastically open to further communication, ensuring thorough resolution to all your questions.

---

### Meta-Review · Area_Chair_h3Ap · 2023-12-15

**Metareview:**

All three reviewers give consistent and positive comments, and the major issues are: (1) Some important details are missing. (2) Show some failure cases. (3) The writing should be improved. After reading the comments, the AC recommends to accept this paper and encourage the authors to take the comments into consideration in their final version.

**Justification For Why Not Higher Score:**

Please see the detailed comments

**Justification For Why Not Lower Score:**

Please see the detailed comments

---

### Decision · Program_Chairs · 2024-01-16

Accept (poster)